# A neuron model with unbalanced synaptic weights explains the asymmetric effects of anaesthesia on the auditory cortex

Luciana López-Jury[1]*, Francisco García-Rosales[1,2], Eugenia González-Palomares[1], Johannes Wetekam[1], Michael Pasek[3], Julio C. Hechavarria[1]*

1 Institute for Cell Biology and Neuroscience, Goethe University, Frankfurt am Main, Germany, 2 Ernst Strüngmann Institute (ESI) for Neuroscience in Cooperation with Max Planck Society, Frankfurt am Main, Germany, 3 Institut für Theoretische Physik, Goethe University, Frankfurt am Main, Germany

* lopezjury@bio.uni-frankfurt.de (LL-J); hechavarria@bio.uni-frankfurt.de (JCH)

## Abstract

Substantial progress in the field of neuroscience has been made from anaesthetized preparations. Ketamine is one of the most used drugs in electrophysiology studies, but how ketamine affects neuronal responses is poorly understood. Here, we used in vivo electrophysiology and computational modelling to study how the auditory cortex of bats responds to vocalisations under anaesthesia and in wakefulness. In wakefulness, acoustic context increases neuronal discrimination of natural sounds. Neuron models predicted that ketamine affects the contextual discrimination of sounds regardless of the type of context heard by the animals (echolocation or communication sounds). However, empirical evidence showed that the predicted effect of ketamine occurs only if the acoustic context consists of low-pitched sounds (e.g., communication calls in bats). Using the empirical data, we updated the naïve models to show that differential effects of ketamine on cortical responses can be mediated by unbalanced changes in the firing rate of feedforward inputs to cortex, and changes in the depression of thalamo-cortical synaptic receptors. Combined, our findings obtained in vivo and in silico reveal the effects and mechanisms by which ketamine affects cortical responses to vocalisations.

**Data Availability Statement:** All data supporting this study can be found in https://gin.g-node.org/Luciana/anesthesia_bats. Underlying codes to

## Introduction

In the past decades, substantial progress has been made in the field of sensory processing of vocalisations. Work on anaesthetized animals has contributed tremendously to the current body of knowledge on how behaviourally relevant sounds are represented in the brain [1–6]. Yet, our understanding of how anaesthetics influence responses to sounds is limited. In this article, we combined computational modelling and in vivo electrophysiological recordings from awake and anaesthetized bats (species: *Carollia perspicillata*) to study the effects of ketamine-xylazine (KX) anaesthesia on the processing of conspecifics vocalisations.

The mixture of KX is widely used in electrophysiological studies in mammals to ensure the absence of pain and the animal's unconsciousness. Interestingly, in the last decade, clinical

generate most of the figures are also available in the same link. DOI for dataset: https://doi.org/10.12751/g-node.4s9mm4

**Funding:** This work was supported by Deutsche Forschungsgemeinschaft (DFG) - Project number 428645493 to JCH. https://gepris.dfg.de/gepris/projekt/428645493 The funders had no role in study design, data collection and analysis, decision to publish, or preparation of the manuscript

**Competing interests:** The authors have declared that no competing interests exist.

**Abbreviations:** AC, auditory cortex; AI, primary auditory cortex; AII, secondary auditory cortex; c.e., context effect; DP, dorsoposterior field; FTC, frequency tuning curve; HF, high-frequency; KX, ketamine-xylazine; LF, low-frequency; RMS, root-mean-square; s.s.s., stimulus-specific suppression.

studies have demonstrated that ketamine is also useful as an antidepressant [7,8]. Ketamine mainly affects glutamatergic transmission inhibiting NMDA receptors [9]. Its effect in the cortex is characterised by increasing low-frequency oscillations and reducing ongoing "spontaneous" activity [10]. However, the effects of KX on stimulus-evoked activity are difficult to generalise across cortical neurons, particularly to complex sounds such as vocalisations [11,12].

Bats represent a good mammalian model to study vocal communication [13,14]. In addition to their own sonar pulses, which enable bats to navigate in the dark, bats are constantly processing echolocation pulses and social calls from conspecifics [15]. These two types of vocalisations differ in terms of their frequency composition: Social calls often have lower fundamental frequencies than echolocation (Fig 1A) [16]. In the auditory cortex of bats, there are specialised neurons that respond well to both types of signals, echolocation and communication, suggesting a multifunction theory of cortical processing [17]. Bat multifunctional neurons have been well characterised in terms of their location within the auditory cortex, responses to pure tones, and even to natural sounds [18–22]. It has been shown that multifunctional neurons increase their discrimination between echolocation and communication calls when the vocalisations are preceded by natural acoustic contexts (Fig 1B) [23]. Multifunctional neurons present a good opportunity to investigate the effects of anaesthesia on sensory processing of vocalisations because (i) they constitute a well-characterised neuronal population; (ii) they are thought to play an important role for communication in complex acoustic environments; and (iii) the synaptic mechanisms underlying the effects of anaesthesia can be studied in silico, since a neuron model that explains responses to individual sounds and sound mixtures in the awake state is now available (Fig 1C) [23].

The main aim of this study was to determine if echolocation and communication acoustic contexts drive similar context-dependent effects in awake and KX-anaesthetized bats. Previous studies have shown that ketamine reduces responses to vocalisations [11,12] and increases adaptation in cortical regions [24,25]. Considering these findings, we modified the neuron model previously constructed for awake data and simulated the expected KX effects. We called this first in silico experiment "naïve modelling". Naïve modelling predicted that known effects of KX would have significant effects on the context-mediated modulation of neural responses, independent on the type of the context that precedes the target sounds (echolocation or communication). Surprisingly, electrophysiological recordings contradicted the naïve model prediction and showed that under KX, context effects were asymmetrical: Only the sound responses after communication context were affected by anaesthesia. A new set of in vivo experiments allowed us to determine that such asymmetries were due to stimulus frequency-specific effects of anaesthesia. The naïve model was updated based on the findings obtained in vivo. From the updated models, the one that succeeded at reproducing all in vivo experimental observations was one in which anaesthesia affected only high-frequency (HF) tuned inputs and their respective synapses, creating a compensation of the effects and causing the apparent absence of effect of anaesthesia after echolocation context.

## Results

### Predicted effects of anaesthesia in auditory cortex (naïve modelling)

We modified the cortical neuron model of [23] to investigate the effects of KX on context-dependent processing of vocalisations. The published model was designed to reproduce spiking responses to acoustic transitions between echolocation and communication sounds in awake bats. It showed that the presence of acoustic context—defined as sequences of echolocation and communication (S1 Fig)—leads to an overall suppression of probe-evoked responses.

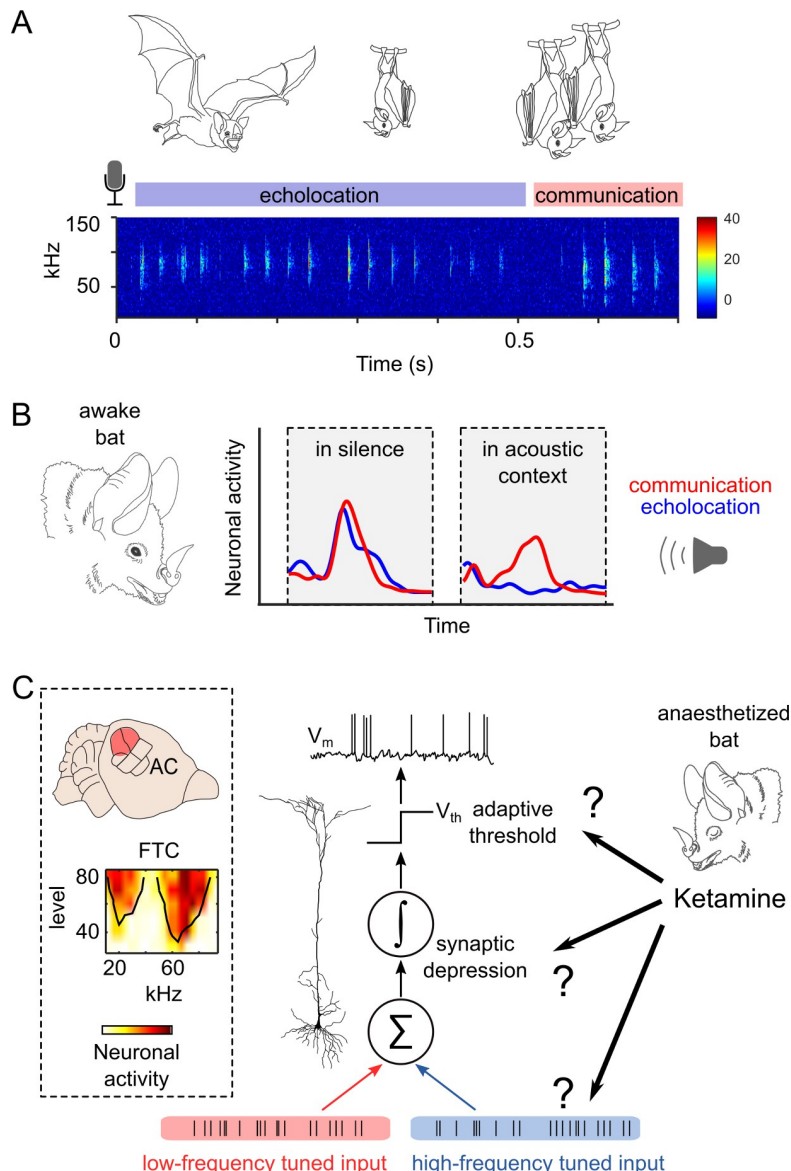

**Fig 1. Effects of anaesthesia on context-dependent processing of vocalisations in the cortex of bats.** (**A**) A representative spectrogram of natural bat vocalisations obtained from a group of bats. The audio recording is composed of a mixture of echolocation pulses and social calls uttered by several conspecifics roosting together. The colour bar indicates sound amplitude (arbitrary units). (**B**) Response of one cortical neuron to playbacks of echolocation and communication calls heard in silence (left) and after acoustic contexts (right). The responses were recorded in the awake state. Data from [23]. (**C**) Top left: Location of the bat auditory cortex (AC). The HF cortical fields that contain multifunctional neurons (such as that represented in B) are highlighted in red. Below: A characteristic multipeaked frequency tuning curve (FTC) from these neurons is shown. Right: An integrate-and-fire neuron model reproduces the context-dependent processing observed experimentally in awake bats. In theory, the known effects of ketamine on evoked responses can be used to predict ketamine effects on context-dependent processing in the bat cortex. Data supporting panel B can be found in "all_psth.mat" and the underlying code in "Fig 1B.m," both in "Ephys data" folder in https://gin.g-node.org/Luciana/anesthesia_bats.

This suppression was found to be stimulus-specific. To study the effects of KX on these findings, we varied model parameters in a manner consistent with experimentally observed effects of KX on evoked responses to complex and natural sounds. For instance, it has been shown

that KX suppresses responses to natural sounds and increases adaptation to repetitive sounds [11,12,24,26,27]. Accordingly, we independently decreased the average firing rate of inputs in response to the context sequences and increased the magnitude of pre- and postsynaptic adaptation, relative to the parameters values fit to the awake data (Fig 2). The mechanisms of neuronal suppression in the model are activity dependent. Therefore, a reduction in the firing rate

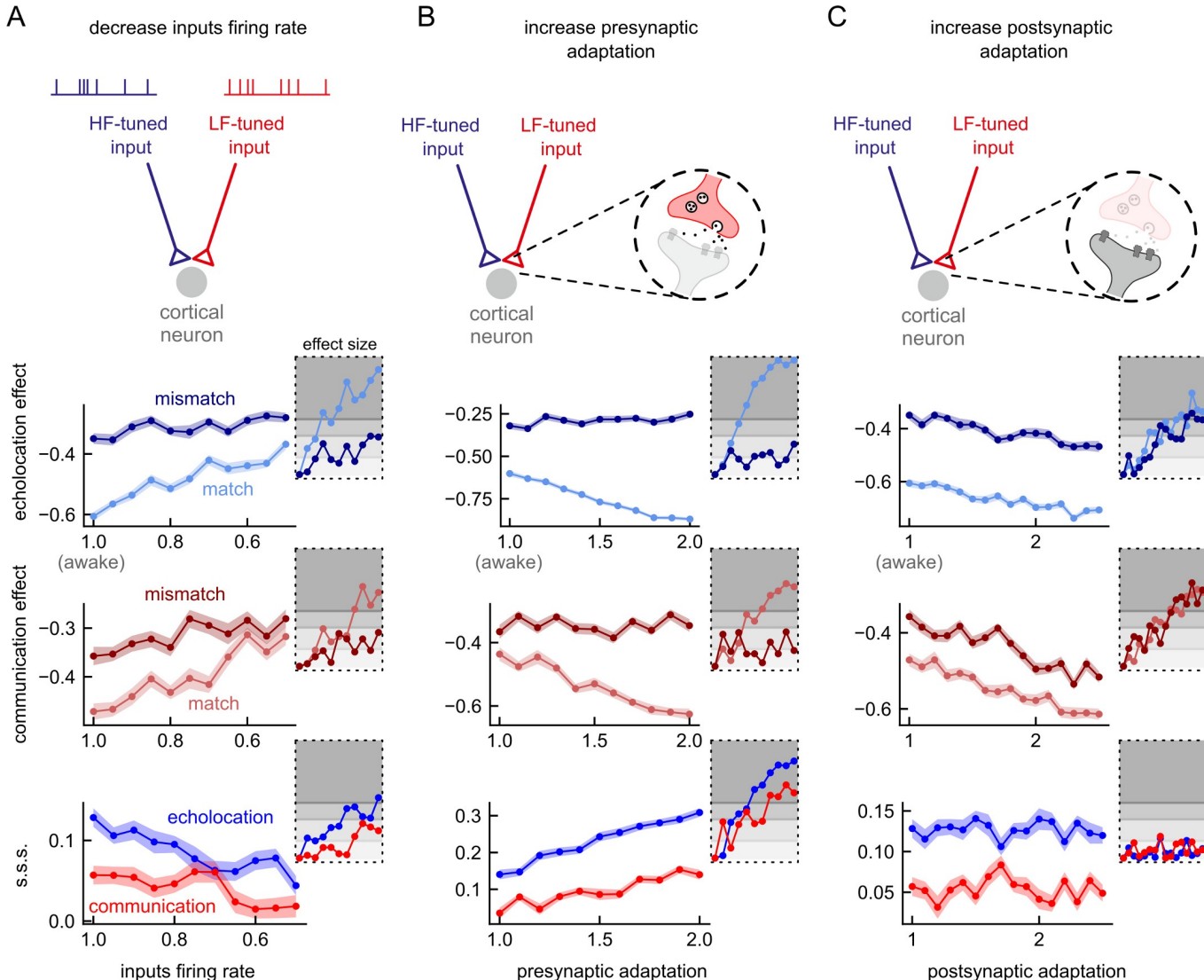

**Fig 2. Effects of anaesthesia on a model for acoustic context effects on cortical neuron responses to natural sounds.** Diagrams on top illustrate possible effects of KX on a neuron model that explains acoustic context effect in the AC of awake bats [23]. (**A**) The decrease of the cortical input firing rate was implemented by systematically decreasing the average input rate parameter, ν, by a factor of 1.0 (awake model) to 0.5 in steps of 0.05. First row shows the context effect after echolocation context for each simulation ($n = 50$) for the mismatching and the matching probes. Second row shows the context effect after communication context. Third row shows the stimulus-specific suppression index (s.s.s.) obtained from the same simulations after echolocation context (in blue) and communication context (in red). Insets indicate the effect size (Cliff's delta) of each simulation for each parameter compared against the respective awake model (i.e., factor = 1.0). The different degrees of grey indicate the different interpretation of effect size: negligible, small, medium, and large. (**B**) The increment of presynaptic adaptation was implemented by systematically increasing the parameter $\Delta_s$, by a factor of 1.0 to 2.0 in steps of 0.1. (**C**) The increment of postsynaptic adaptation was implemented by systematically increasing the adaptive threshold time constant, $\tau_{th}$, by a factor of 1.0 to 2.5 in steps of 0.1. Note that presynaptic adaptation accentuated differences between probe-responses, while postsynaptic adaptation left them unchanged. Data supporting this figure can be found in "coefC_tau_th.npy" and "d_tau_th.npy" in "Model" folder in https://gin.g-node.org/Luciana/anesthesia_bats. The underlying codes for each panel are in file "plots.py" in the same location. LF, low-frequency; HF, high-frequency.

at the input level decreased the suppression on both probe responses (context effect values closer to zero; Fig 2A top, middle). However, the effect size was largest in the matching probes (e.g., communication probe following communication context). Consequently, the stimulus-specific suppression (s.s.s.) index, defined as the difference between context effects on matching and mismatching probes, decreased (Fig 2A bottom). On the other hand, an increment of adaptation had the opposite outcome: It increased context effect on matching probes, in the case of presynaptic adaptation (Fig 2B top, middle), and it increased context effect on both probes, in the case of postsynaptic adaptation (Fig 2C top, middle). Due to its observed unbalanced effect on probe responses, increasing presynaptic adaptation led to an increment of s.s.s. (Fig 2B bottom), while the balanced effect of postsynaptic adaptation had no repercussions on s.s.s. (Fig 2C bottom).

As expected, all the above models of anaesthesia showed a decrement in the spiking activity in response to natural sounds compared to the awake model. In general, the size of the effects was similar between echolocation and communication or slightly larger for echolocation context rather than communication (insets in Fig 2).

Next, we ran several simulations covarying two parameters to determine how the interactions between two effects of anaesthesia influence the outcome of the models in comparison to the awake model (Figs 3 and S2). Considering that the reduction of inputs' firing rate and the enhancement of adaptation have opposite consequences on the context effects in the model, we hypothesised that they could compensate each other and lead to an unchanged outcome. Indeed, the results of the simulations where we decreased input activity as well as increased presynaptic adaptation showed a region of parameters wherein changes in the context effect on matching probes and in the s.s.s. was negligible relative to the awake model (Fig 3A, pale diagonal in first and third column). In the same simulations, the effect of the combined decreasing input activity and increasing presynaptic adaptation in the case of a mismatching probe was generally a reduction of the context effect compared to the awake state (corresponding to positive values of Δ c.e.; Fig 3A middle column). Compensation of effects was also observed between inputs activity and postsynaptic adaptation (S2A Fig). However, the compensatory region of parameters with negligible changes appeared for both probes, matching and mismatching. As expected, the increment of both types of adaptations, pre- and postsynaptic, generated increment on context effects; stronger suppression was observed on the matching probe (compared to the mismatching one) for all the range of parameters tested. In addition, the s.s.s. increased after both contexts due to presynaptic adaptation increment (S2B Fig).

To better visualise the interaction between input firing rate and adaptation, we plotted the time course of presynaptic adaptation and the respective spiking activity for each input and the cortical neurons. We compared the results for three different models during communication context simulations (Fig 3B). In all the models, the synaptic strength associated with low-frequency (LF) inputs decreased throughout the context due to presynaptic adaptation. In the awake model, at the end of the context, the synaptic strength was approximately 0.6 (Fig 3B left column). This value depends on the total amount of spikes evoked by the context as well as on the amount of presynaptic depression per spike. In a model in which the amount of presynaptic depression per spike is higher, but the input activity is constant, the synaptic strength reached even lower values at the end of the context, approximately 0.2 (Fig 3B middle column). Finally, decreasing the spiking of the inputs in the preceding model compensated the effect of a strong spike-dependent depression by reducing the total amount of spikes in the inputs (Fig 3B right column). The strength of the synapse at the end of the context is then similar in magnitude to that observed in the awake model (approximately 0.6).

To summarise this section, modifying a model based on awake data to simulate the effects of KX on neuronal activity resulted in (i) decreased spiking in the cortical neurons

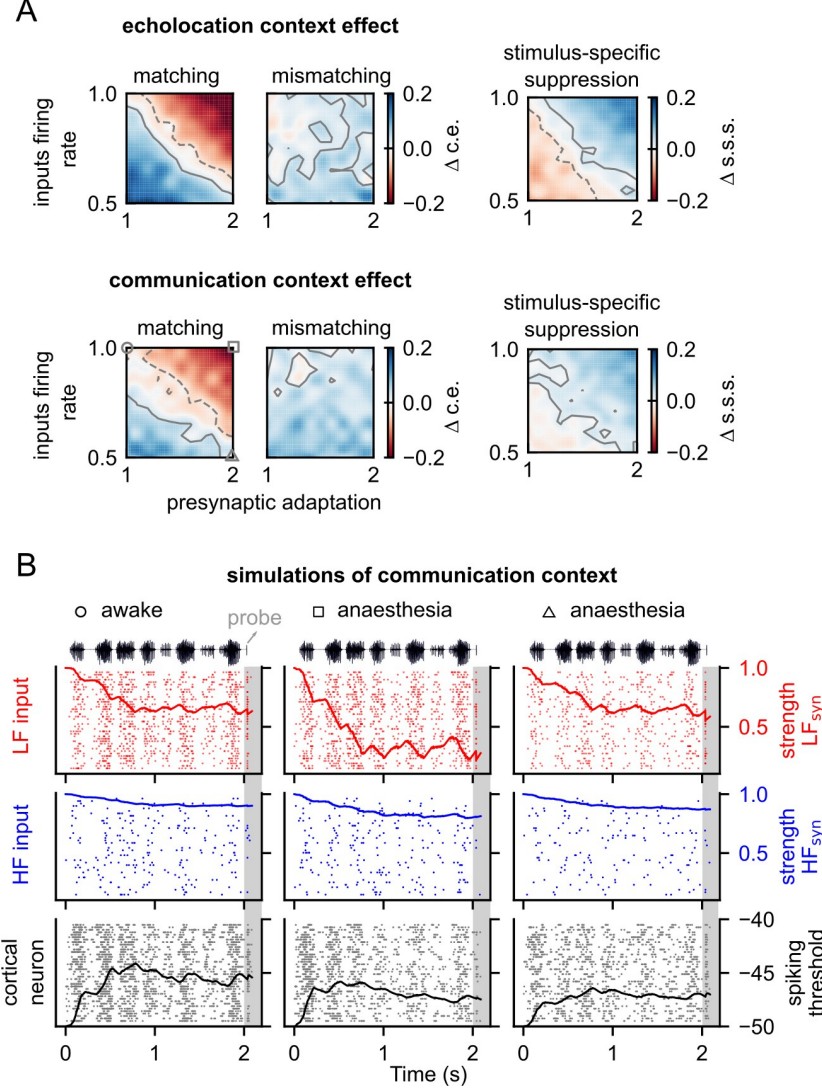

**Fig 3. Interaction between two anaesthesia effects leads to a compensatory mechanism in the outcome of the model.** (**A**) Difference between context effect, c.e. (left) and stimulus-specific suppression, s.s.s. (right) when decreasing inputs firing rate and increasing presynaptic adaptation relative to the respective value obtained with the awake model. First row, after echolocation. Second row, after communication context. Contour lines in grey indicate the area with negligible effect size. Positive contours are indicated by solid lines and negative, by dashed lines. The anaesthesia effects were implemented as described in Fig 2. Note that red colours in the colormaps of c. e. correspond to an increment of suppression, and blue, a decrement. (**B**) Raster plots of 30 trials for the LF tuned input (top), HF tuned input (middle), and cortical neuron (bottom) during a simulation of communication context and matching probe for three combination of parameters: (1) no anaesthesia (circle), which corresponds to inputs firing rate factor = 1.0 and presynaptic adaptation factor = 1.0, this represents no change from the awake model; (2) under anaesthesia (square) was obtained with double of presynaptic adaptation used in the awake state; and (3) under anaesthesia (triangle) was obtained with double presynaptic adaptation but half of inputs firing rate. Lines on the top rasters indicate the average strength of the synapse ($X_s$) between LF tuned inputs and cortical neurons along time. Lines on the middle rasters indicate the same but from the HF tuned inputs. Lines on bottom rasters indicate the average spiking threshold ($\omega_{th}$) of the cortical neuron across trials along time. Data supporting panel A can be found in "coefC_d.npy" in "Model" folder in https://gin.g-node.org/Luciana/anesthesia_bats. The underlying code is in the file "plots.py" in the same location.

independently of the type of acoustic context to which the animals were subjected; (ii) decreased context effect when KX affects spiking activity of the cortical inputs; and (iii)

increased context effect when KX affects neural adaptation. It is important to note that responses to both contexts, echolocation and communication, exhibited very similar patterns of change under the simulated effects of anaesthesia.

## Predicted effects of anaesthesia in a cortical circuit that includes inhibition (naïve modelling)

It has been shown that ketamine can cause cortical disinhibition through selective antagonist effects of inhibitory interneurons [28]. Therefore, we decided to include interneurons in our cortical model in order to study how anaesthesia may affect excitatory neurons in a circuit that includes inhibition. The new circuit consists of our previously established model of an excitatory pyramidal neuron [23] reciprocally connected with an inhibitory interneuron (Fig 4A). After setting the synaptic weights of the circuit to reproduce the spiking of the neurons in response to the calls in silence observed in vivo, the pyramidal neuron model was still able to exhibit s.s.s. in acoustic contexts (Fig 4B). The antagonist action of ketamine on the interneurons' NMDA receptors was modelled by decreasing the synaptic weight associated with the excitatory inputs of interneurons ($w_{e,i}$). The simulations showed no effect on the s.s.s. (Fig 4C top). Because ketamine affects glutamatergic transmission in general, we also tested the effects on s.s.s. when decreasing the synaptic weight associated with the excitatory synapses in pyramidal neurons. These simulations did not show any effect compared to the awake model (Fig 4C bottom).

Next, we ran simulations in which we decreased both synaptic weights simultaneously. The firing rate of the pyramidal neurons in response to both contexts is plotted in Fig 4D for all the simulations. As expected, if the decrement of the synaptic weight associated with interneurons ($w_{e,i}$) is higher than the one associated with pyramidal neurons ($w_{e,e}$), the spiking of the pyramidal neurons increased relative to the awake model (blue colour). This agrees with the disinhibition hypothesis of ketamine's action [29,30]. From all the anaesthetized models tested so far (Figs 2, 3 and S2), this was the only model able to increase excitatory synaptic drive in cortical neurons following ketamine injection. From the same simulations, we calculated the s.s.s. and plotted the difference with the awake model. There was no clear pattern of effect on the s.s.s. index neither after echolocation nor after communication contexts (Fig 4E).

Because there is evidence that the afferents of interneurons can also be long range [31], we implemented an alternative model in which inhibitory neurons and pyramidal neurons share common inputs (S3 and S4 Figs). We dismissed models in which inhibitory neurons were selective for one of the two frequency-bands of the inputs' tuning because of two reasons. One, it has been shown that inhibitory neurons are often more broadly tuned than excitatory neurons [32], or in some cases, just balanced [33]. The second reason is due to the fact that selective interneurons affect the selectivity of pyramidal neurons (S3A and S3B Figs) and the neurons targeted in this study are precisely nonselective for communication and echolocation sounds in silent contexts, as illustrated in Fig 1B. A model whose inputs to interneurons are balanced and therefore present no preference for any call in silence (S3C and S4A Figs) exhibited s.s.s. after both acoustic contexts in the awake state (S4B Fig). When testing the effects of anaesthesia on this model (S4C Fig), we obtained similar results to those with the reciprocally connected model. The main observation was an increment of firing rate in response to the context sequences when anaesthesia affects more strongly the synapses of inhibitory neurons than excitatory (S4D Fig). We observed no effects on the s.s.s. when decreasing the synaptic weights of either of the two excitatory synapses (S4E Fig).

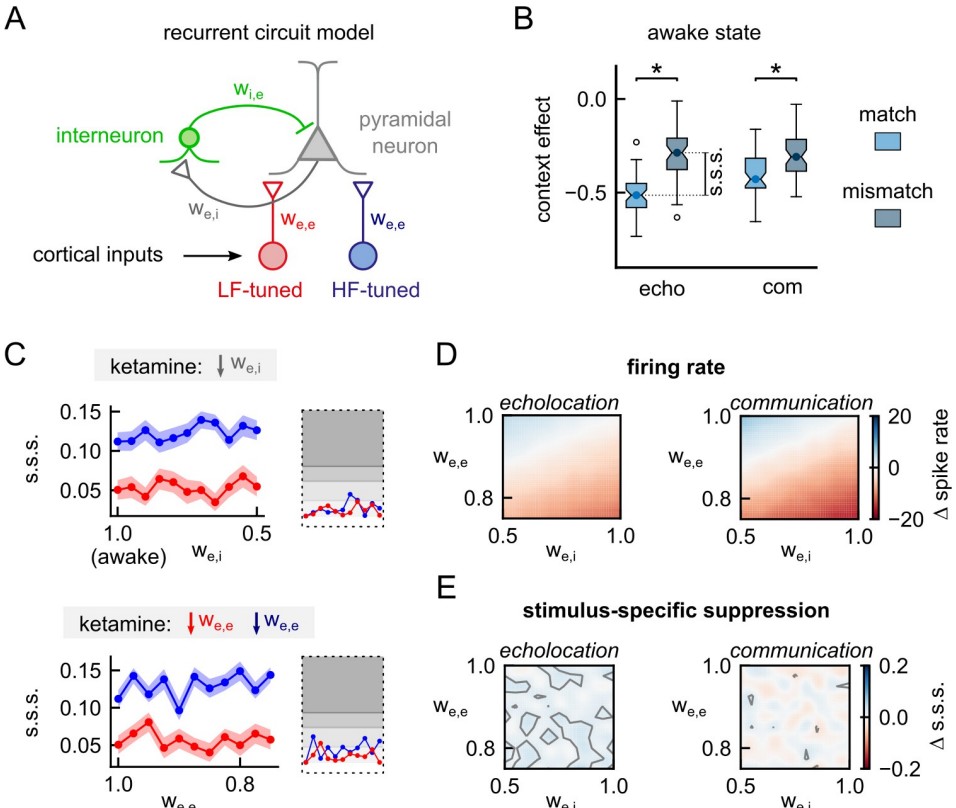

**Fig 4. Effects of anaesthesia on context-dependent processing in a cortical circuit model with interneurons.** (**A**) Diagram of the circuit model. (**B**) Context effect on probe responses obtained from a model as illustrated at the left simulating an awake state. (**C**) Top: Stimulus-specific suppression index (s.s.s.) obtained from simulations in which the synaptic weight of the excitatory input to the interneuron ($w_{e,i}$) was systematically reduced by a factor of 1.0 (awake model) to 0.5 in steps of 0.05. In blue, the s.s.s. obtained after an echolocation context; in red, after communication context. Inset indicates the effect size (Cliff's delta) of each simulation compared against the awake model. The degrees of grey indicate the standard interpretation of the effect size: negligible, small, medium, and large. Bottom: Same than above, but s.s.s. were obtained from stimulations in which the synaptic weight of the excitatory inputs to the pyramidal neuron ($w_{e,e}$) were systematically reduced by a factor of 1.0 to 0.75 in steps of 0.025. (**D**) Combined effect of the reduction of excitatory inputs to the pyramidal neuron and to the interneuron on the firing rate of the pyramidal neurons in response to the sequences used as contexts. (**E**) Same simulations than above but showing the variation in the s.s.s. in relation to the awake model. Data supporting panels B-E can be found in "w1-w2_recurrentv2.npy" in "Model" folder in https://gin.g-node.org/Luciana/anesthesia_bats. The underlying codes are in "plots.py" in the same location.

## Effects of anaesthesia in in vivo preparations

To experimentally assess the effect of KX on context-dependent processing of vocalisations, we performed single-neuron recordings in the AC of anaesthetized bats and compared the neuronal responses to the same paradigm of stimulation used in a previous study in awake animals [23]. Our goal was to validate one of the model predictions and, thus, determine the mechanisms underlying the KX effects on evoked responses to natural sounds. We recorded a total of 107 units from HF fields of the AC under anaesthesia. As expected, several neurons presented multi-peaked frequency tuning curves ($n = 62$; S5B–S5D Fig). This tuning shape has been linked to HF areas in *C. perspicillata* in previous studies [34,35]. Here, we focus our analysis only in a subset of neurons characterised electrophysiologically as being "equally responsive" to both sound categories: an echolocation pulse and a communication syllable ($n = 46$; S5C Fig). This classification was made following the same criteria used to describe context effects in awake bats [23].

The awake and anaesthetized preparations presented similar physiological properties, such as tuning to pure tones (iso-level frequency tuning curves) and responses to natural sounds (S5D–S5F Fig). Two representative examples from each preparation are given in Fig 5. In both cases, neurons classified as "equally responsive" to both probe sounds in the absence of context exhibited strongest responses to mismatching probe sounds presented 60 ms after the end of the echolocation context (Fig 5 top). However, a different outcome was observed after the communication context. Although the response to the mismatching probe was stronger than to the matching probe in the awake example neuron, the responses under anaesthesia were strongly suppressed for both matching and mismatching probes (Fig 5 bottom). Only the communication context under anaesthesia presented a negative value of s.s.s., indicating that the response to the probe after context was more suppressed for the mismatching probe than for the matching probe (suppression calculated relative to responses obtained in the absence of context).

While the majority of neurons recorded from awake bats showed positive s.s.s. after echolocation and communication contexts, neuronal responses under anaesthesia exhibited positive values only after echolocation context (Fig 6A and 6B). The average s.s.s. obtained after echolocation was higher than that for communication in both preparations (Fig 6C, paired test

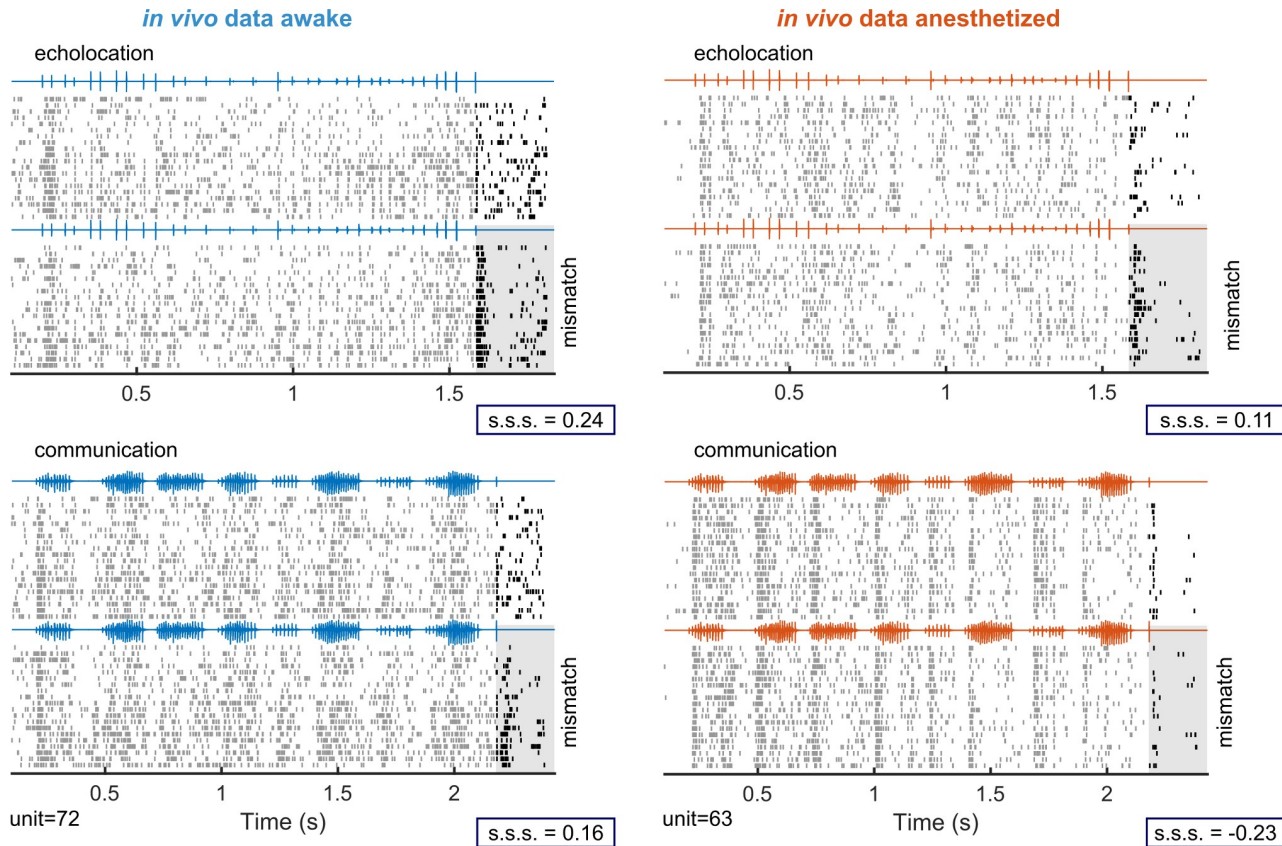

**Fig 5. Examples of effects of anaesthesia on context-dependent processing in cortical neurons.** Two example units illustrating cortical responses to natural sounds in awake (left) and anaesthetized (right) bats. Top: spiking responses to a sequence of echolocation pulses (context) followed by a matching probe sound (top 20 trials) and by a mismatching probe sound (bottom 20 trials). After a context sequence of echolocation pulses, matching and mismatching probe sounds correspond to a single echolocation pulse and a communication call, respectively. Bottom: the responses of the same neurons to a communication sequence followed by matching and mismatching probe sounds, i.e., a single communication call and by an echolocation pulse, respectively. The grey shaded area is used to highlight the response to the mismatching probe sound in both contexts. For each example unit and context, the respective value of stimulus-specific suppression ("s.s.s.") is indicated.

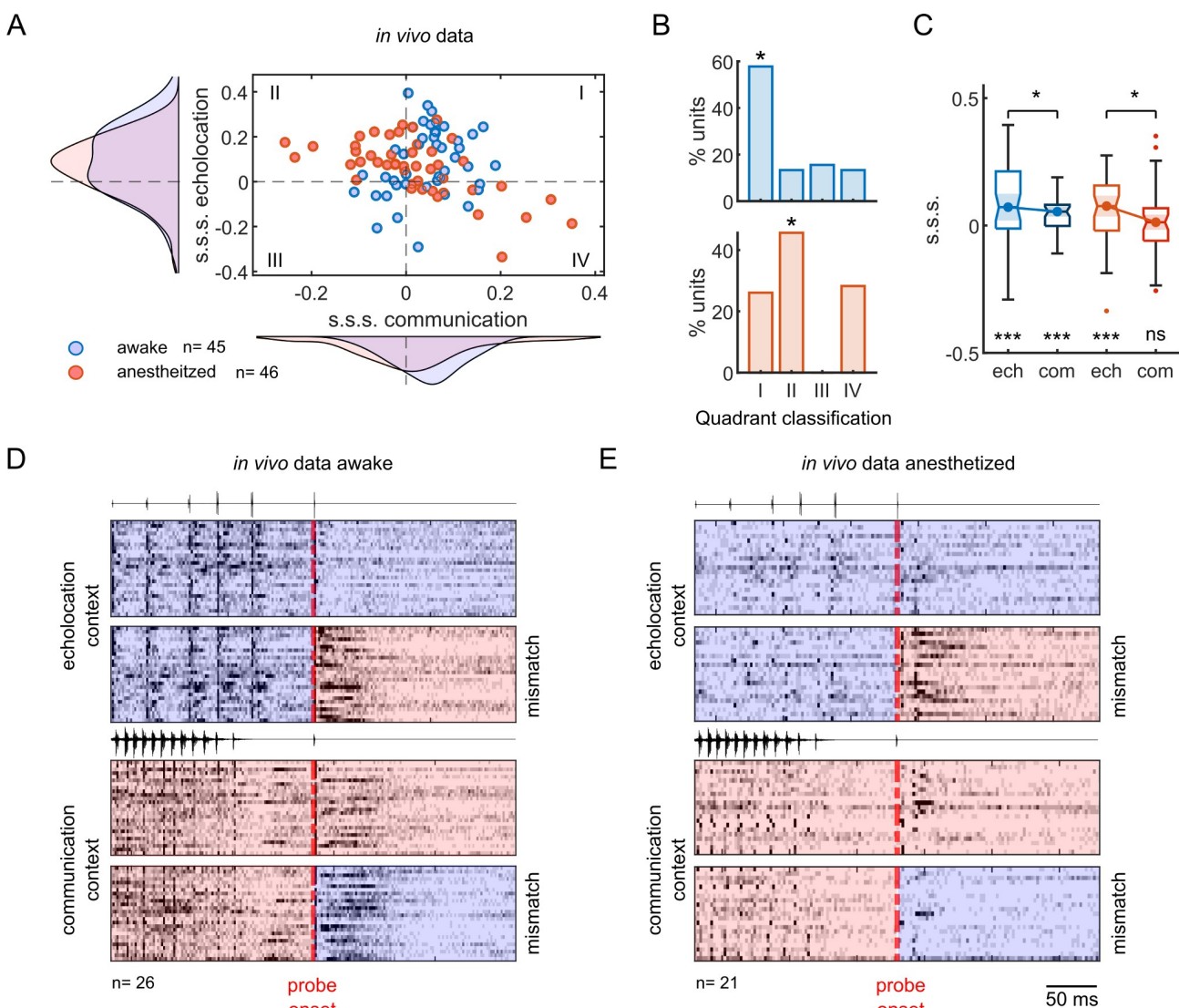

**Fig 6. Anaesthesia decreases stimulus-specific suppression on probe-responses only after communication context.** (**A**) Stimulus-specific suppression indexes (s.s.s.) calculated from awake (blue) and anaesthetized (red) preparations. Each circle corresponds to one unit ($n = 45$ awake, $n = 46$ anaesthetized). The curves represent the distribution for a particular context. (**B**) Percentage of units per quadrant showed in "A" of units recorded in awake (top) and anaesthetized animals (bottom). Asterisks show the most represented quadrant on each preparation. (**C**) Boxplots of s.s.s. per context in awake and anaesthetized preparations. Significance level below the boxplot indicates differences against null distribution (one-sample Wilcoxon signed-rank test). Above, the comparison between contexts (paired test Wilcoxon signed-rank). (**D**) Normalised and averaged firing rate of units that corresponded to the most representative categories in "B" (quadrant I for awake and quadrant II for anaesthetized) aligned by probe onset. Blue background corresponds to echolocation sounds stimulation and red, to communication, either context or probe. (**E**) Same than in "D" but for the anaesthetized preparation. All the results correspond to a gap between context offset and probe onset equal to 60 ms. Data supporting the panels A-C can be found in the files "all_data.mat," "pt_data.mat," "all_dd.mat," and "all_ce.mat" in "Ephys data" folder in https://gin.g-node.org/Luciana/anesthesia_bats. The underlying codes for each panel are in the same location.

Wilcoxon signed-rank, $p = 0.046$ in awake, $p = 0.049$ in anaesthetized). However, only after communication context under anaesthesia was the s.s.s. not significantly different from a distribution centred on zero, while the rest of the distributions were all significantly above zero (Fig 6C, one-sample Wilcoxon signed-rank test, $p = 6.3 \times 10^{-4}$ for echolocation-awake, $p = 1.7 \times 10^{-4}$ for communication-awake, $p = 7.2 \times 10^{-4}$ for echolocation-anaesthetized, $p = 0.4$ for communication-anaesthetized). To visualise the spiking responses in time under

both states, we plotted the instantaneous firing rate aligned to the probe onset for neurons that corresponded to the most represented category (Fig 6B) in awake and anaesthetized preparations. Although neuronal responses to echolocation and communication probe sounds in isolation were comparable in both states (S6 Fig), after echolocation context, neurons exhibited a clear preference for the mismatching sound, i.e., communication probe (Fig 6D and 6E top). Regarding activity during the echolocation context, the responses under anaesthesia were lower in comparison to those in awake state. In awake animals, although there were differences in the responses to the two probes after communication context, the preference for the mismatching probe was less clear than after echolocation (Fig 6D bottom). On the other hand, for units recorded under anaesthesia, there was no difference in the magnitude of the responses to the probes, both being strongly suppressed after context independently of the probe type (Fig 6E bottom).

In general, in both preparations, acoustic context suppressed the responses to the probe sounds (negative values of context effect; Fig 7A and 7B). In agreement with previous results quantifying s.s.s., the context effects were significantly different between probes in both states after echolocation context (Fig 7A; Wilcoxon signed-rank test, $p = 6.3 \times 10^{-4}$ in awake, $p = 7.2 \times 10^{-4}$ in anaesthetized) and, only in awake state, after communication (Fig 7B; Wilcoxon signed-rank test, $p = 1.7 \times 10^{-4}$ in awake, $p = 0.45$ in anaesthetized). Using a nonpaired statistical test, we compared the context effect between preparations. There were only significant differences between communication context effects. The context effect decreased for both matching and mismatching probes under anaesthesia relative to the awake treatment (Fig 7A and 7B; Wilcoxon rank-sum test, $p = 0.33$ for echolocation-matching, $p = 0.99$ for echolocation-mismatching and $p = 0.02$ for communication-matching, $p = 0.006$ for communication-mismatching).

Other features such as spontaneous firing rate and evoked response by the context sequences were also compared across preparations (Fig 7C). The average spontaneous firing rate of the neurons was lower in the preparation under anaesthesia compared to awake (Fig 7C; Wilcoxon rank-sum test, $p = 1.7 \times 10^{-4}$). Similarly, the evoked response during the echolocation context also decreased (Fig 7C; Wilcoxon rank-sum test, $p = 0.0013$). However, no significant differences were found in the evoked responses during the communication context between preparations (Fig 7C; Wilcoxon rank-sum test, $p = 0.17$).

To corroborate the comparisons between preparations, we performed new experiments testing the effect of KX directly on the same neurons. The neurons were recorded both prior to and following anaesthesia administration. Although we recorded a reduced sample size ($n = 7$), as it was difficult to hold the same neurons for long time periods and only one neuron could be recorded in each animal, the trend already showed the same effects obtained from different preparations (S7 Fig). The effect size between context effects calculated before and after the injection of KX was, on both probe sounds, lower after echolocation than communication context (S7A Fig; Cliff's delta, $d = 0.14$ for echolocation-matching, $d = 0.18$ for echolocation-mismatching, $d = 0.30$ for communication-matching, and $d = 0.32$ for communication-mismatching). In addition, the same neurons showed a significant decrease in the spontaneous firing rate (S7B Fig; Wilcoxon signed-rank test, $p = 0.01$) and in the evoked response only during echolocation context (S7B Fig; Wilcoxon signed-rank test, $p = 0.01$ for echolocation and $p = 0.15$ for communication) after the KX injection. Altogether, these results replicated those obtained between different preparations.

All the results presented so far have been obtained using context-probe stimuli with a time interval of 60 ms between the end of the context and the onset of the probe. In order to explore the temporal course of the context effects, we also used a gap of 416 ms during the recordings under anaesthesia in a subset of neurons (38 out of 46). The same gap was used to stimulate

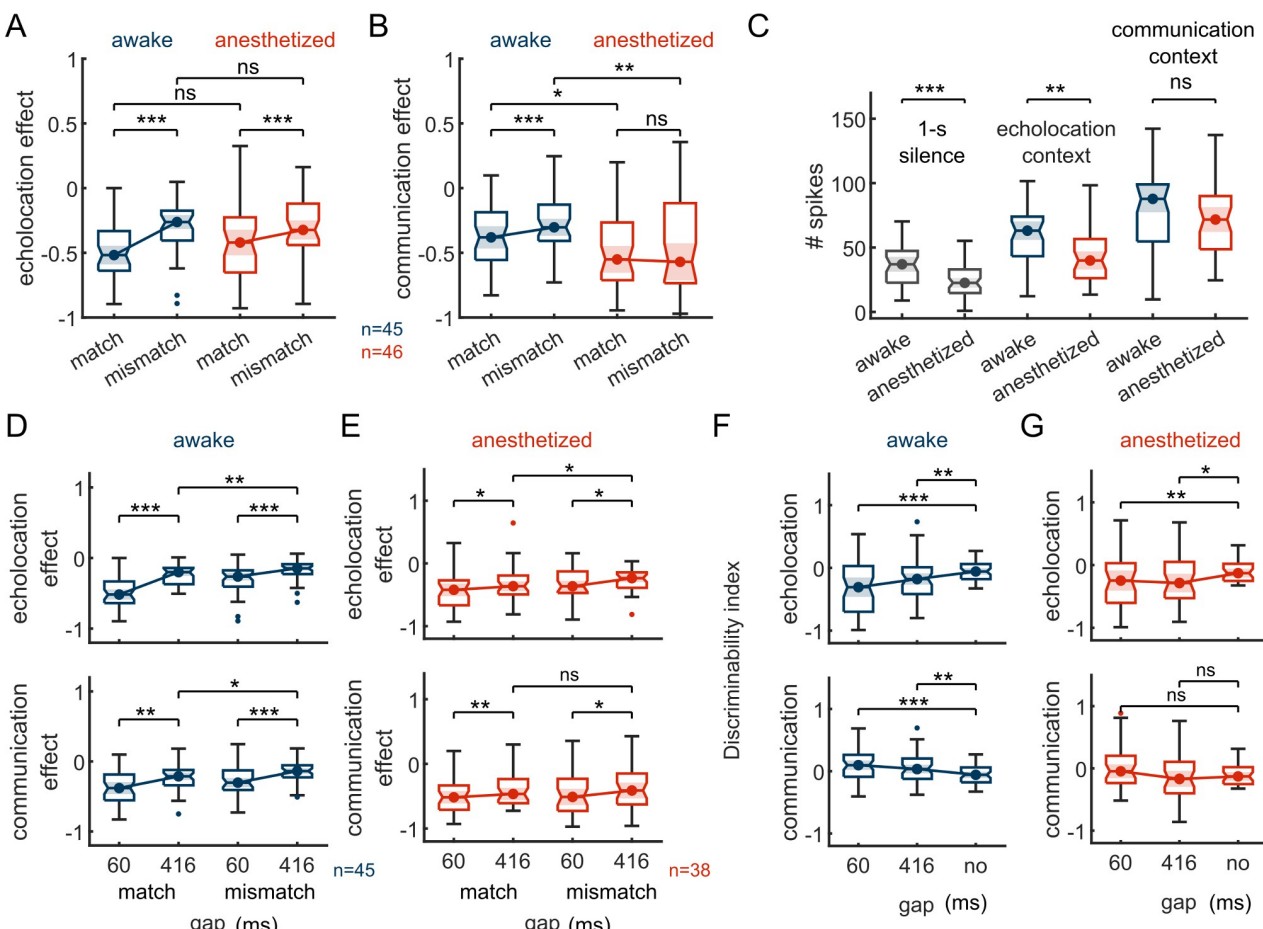

**Fig 7. Stronger suppression on both probes after communication context explains decrement on s.s.s. under anaesthesia.** (**A**) Context effect on probe-responses following an echolocation sequence. In blue, neurons recorded from awake bats ($n = 45$); in red, from anaesthetized ($n = 46$). (**B**) Context effect after communication context. (**C**) Spike counts during 1 second of silence, the echolocation context, and the communication context; for awake and anaesthetized preparations. (**D**) Top: comparisons between context effect after echolocation context with a gap of 60 ms and 416 ms between probe onset and context offset in awake animals. Bottom: same than above, but after communication context. (**E**) Same than in "D" but in anaesthetized animals. (**F**) Discriminability index calculated after 60 and 416 ms from the offset of the echolocation (top) and communication context (bottom) and after no context in the awake preparation ($n = 45$). (**G**) Same than in "F" but the neurons were recorded in anaesthetized bats ($n = 38$). Statistical tests within preparations were obtained using a paired test (Wilcoxon signed-rank). Statistical differences between awake and anaesthetized preparations were obtained using Wilcoxon rank-sum test. Data supporting this figure can be found in the files "all_data.mat," "inj_data.mat," "chi_data.mat," "sp_context.mat," and "all_ce.mat" in "Ephys data" folder in https://gin.g-node.org/Luciana/anesthesia_bats. The underlying codes for each panel are in the same location.

the neurons in the awake preparation [23]. In both preparations, the context effect showed significant reduction after a gap of 416 ms compared to 60 ms (Fig 7D and 7E; Wilcoxon signed-rank test, $p = 6.7 \times 10^{-8}$, $p = 2.0 \times 10^{-5}$, $p = 0.001$, $p = 3.8 \times 10^{-5}$ for echolocation-matching, echolocation-mismatching, communication-matching, and communication-mismatching, respectively, for awake, and $p = 0.02$, $p = 0.03$, $p = 0.002$, $p = 0.03$ for anaesthetized). Despite the decay in context effects observed at longer gaps, the effect after echolocation context was still stimulus-specific, i.e., there were significant differences in context effect in matching versus mismatching probes in both states (Fig 7D and 7E top; Wilcoxon signed-rank test, $p = 0.007$ for awake and $p = 0.02$ for anaesthetized). On the other hand, communication context effect was stimulus-specific for 416-ms gaps only in the awake preparation (Fig 7D and 7E bottom; Wilcoxon signed-rank test, $p = 0.02$ for awake and $p = 0.7$ for anaesthetized). This was

expected considering that the effect was not stimulus-specific even 60 ms after the end of the communication sequence (see Fig 7B). It is worth mentioning that the effect size (Cliff's delta, $d$) of the context effects between gaps was always smaller under anaesthesia than in the awake condition ($d = 0.46$ for echolocation and $d = 0.38$ for communication in awake preparation; $d = 0.16$ for echolocation and $d = 0.11$ for communication in anaesthetized preparation). This result is indicative of a slower recovery of the context effects under anaesthesia.

From the experiments with awake animals, the neurons increased their discriminability for the probe sounds after both contexts and for both gaps compared to when there was no context (Fig 7F; Wilcoxon signed-rank test, $p = 7.6 \times 10^{-5}$ for echolocation-60 ms, $p = 0.001$ for echolocation-416 ms, $p = 1.9 \times 10^{-4}$ for communication-60 ms, $p = 0.002$ for communicacion-416 ms). Here, we showed the same occurred under anaesthesia, but only after echolocation context (Fig 7G; Wilcoxon signed-rank test, $p = 0.001$ for echolocation-60 ms, $p = 0.01$ for echolocation-416 ms, $p = 0.2$ for communication-60 ms, $p = 0.6$ for communicacion-416 ms).

## Asymmetric effects of anaesthesia on the processing of echolocation versus communication sounds is linked to frequency-specific effects

Anaesthesia influenced asymmetrically the cortical processing of echolocation and communication calls: KX modulated the context effects on the probe responses only after the communication context. However, the neuronal responses during the context were only affected for echolocation (summary in Fig 8 top). None of the models presented at the beginning of this article (Figs 2–4) predicted such asymmetrical modulation. Our results indicate that the known effects of KX implemented in our anaesthesia models cannot explain the experimental data obtained in vivo. We conducted several follow up in vivo and modelling experiments to disentangle the possible causes of these unexpected results.

First, we noted that the echolocation and communication sequences used as context differ in their temporal pattern and frequency composition. Therefore, we reasoned two possible scenarios that could explain the observed asymmetries: (i) anaesthesia affects the general neuronal dynamics in a way that only have consequences in the firing rate associated with one context due to its particular temporal modulation (Fig 8A); or (ii) anaesthesia affects the synaptic inputs associated only with one frequency channel (HF, for example), and, therefore, it has repercussions in one context due to its carrier frequency (Fig 8B).

To determine if the asymmetries arose due to differences in the temporal patterns or to the frequency composition of the contexts, we performed a new set of in vivo experiments. We built artificial context sequences combining aspects of the original sequences. The new "chimera" sequences included (i) a "fast echolocation" context, which consisted in several echolocation pulses following the temporal modulation of the original communication context (Fig 9A left); and (ii) a "slow communication" in which a sequence of communication syllables followed the temporal pattern of the original echolocation context (Fig 9A right). The probes and the gaps remained unchanged.

We recorded 64 neurons under anaesthesia in response to the artificial context-probe sequences. A total of 26 neurons satisfied the same electrophysiological criterion of being "equally responsive" to both probe sounds in silence. An example is showed in Fig 9A. The neuron showed preference for the mismatching sound after a sequence of echolocation pulses that doubled the original call rate. However, the neuron did not present a preference for any of the probes after a "slow communication" context.

At the population level, despite the modification of temporal patterns of the calls, the results were similar to those obtained with the natural sequences under anaesthesia. We observed a stronger suppression on the matching probe responses only after "fast echolocation" (Fig 9B;

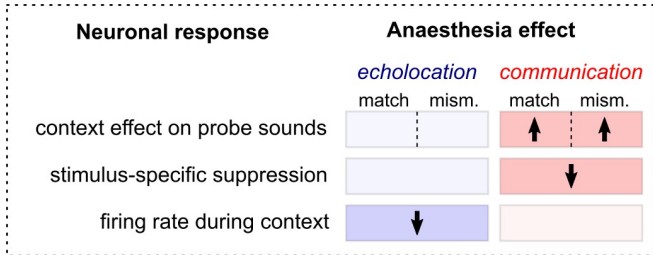

A **Temporal pattern of the contexts**

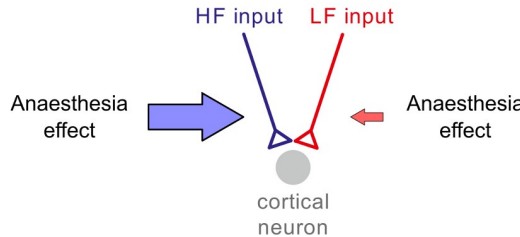

**Fig 8. Anaesthesia modulates neuronal responses in an asymmetric manner for echolocation and communication context-probe paradigm.** At the top, a summary of the experimental results. Direction of the significant changes is indicated with arrows on three different results regarding context-probe paradigm. Note that the changes occur in an asymmetrically between contexts, which can be explained either by (**A**) differences in the temporal pattern of the contexts or (**B**) differences in the frequency composition of the contexts.

Wilcoxon signed-rank test, $p = 3.9 \times 10^{-4}$ for fast echolocation and $p = 0.9$ for slow communication). The "slow communication" context broadly suppressed the responses to both probes, independently on the type of probe. In agreement with this, the s.s.s. was higher for "fast echolocation" in comparison with "slow communication" (Fig 9C top; Wilcoxon signed-rank test, $p = 0.03$), and the latter exhibited a distribution centred at zero (Fig 9C top; one-sample Wilcoxon signed-rank test, $p = 3.9 \times 10^{-4}$ for fast echolocation and $p = 0.9$ for slow communication). The majority of the neurons presented a positive s.s.s. for "fast echolocation," which corresponded to the quadrants I and II (Fig 9C bottom), based on the neuronal classification in the s.s. coordinates plot (Fig 6A). In terms of neuronal discriminability between the probes, the "fast echolocation" context increased the discriminability index obtained after both gaps, 60 and 416 ms, compared to the index calculated in isolation (Fig 9D top; Wilcoxon signed-rank test, $p = 0.04$ for 60-ms gap and $p = 0.009$ for 416-ms gap). On the contrary, the "slow communication" context did not increase discriminability significantly, neither at gaps of 60 ms nor 416 ms (Fig 9D bottom; Wilcoxon signed-rank test, $p = 0.15$ for 60-ms gap and $p = 0.6$ for 416-ms gap).

To directly test the effect of temporal modulation on the context effects, we performed neuronal recordings in response to both types of context, natural and artificial, in the same

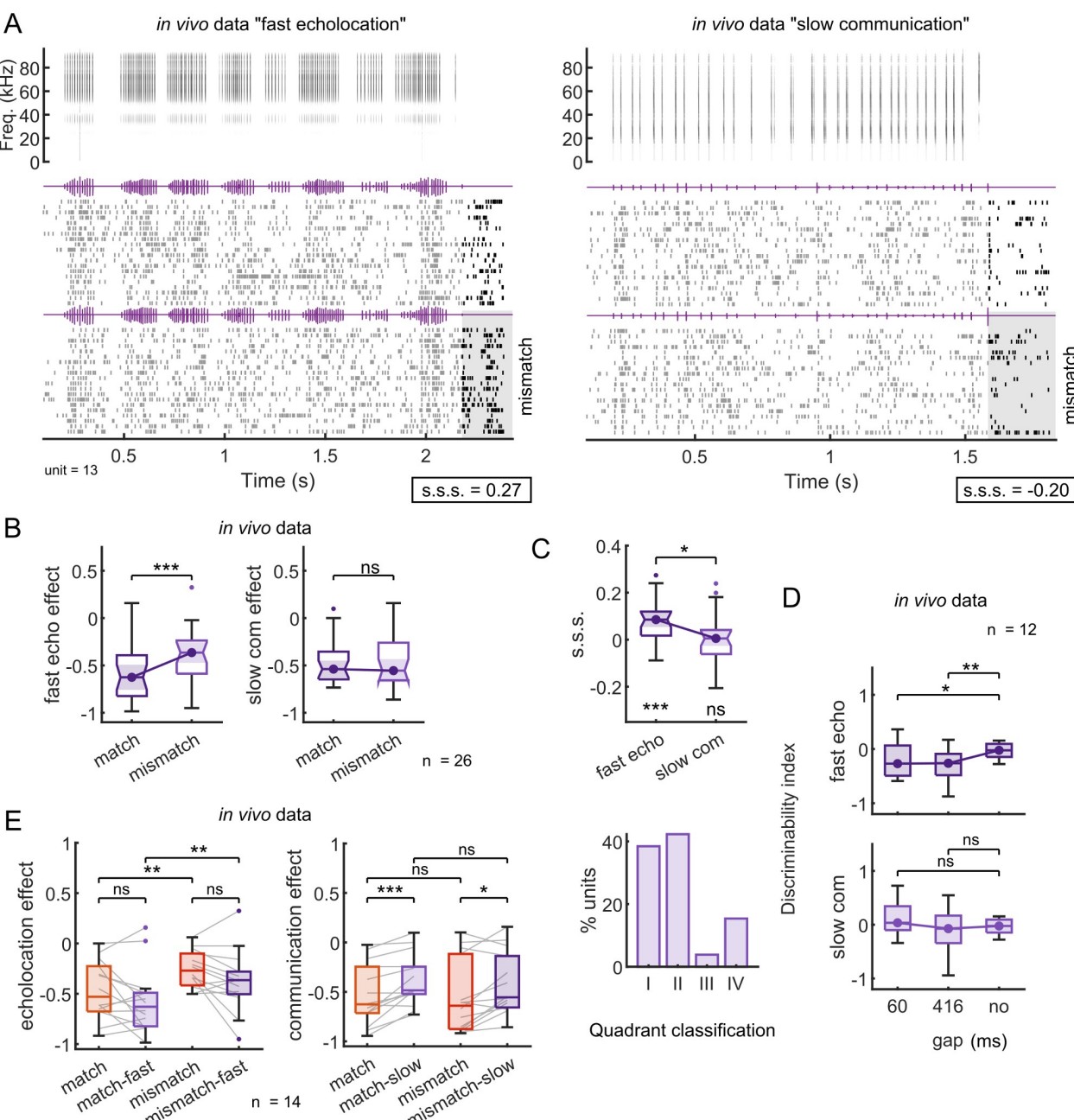

**Fig 9. Switching the temporal patterns between contexts does not change the effect on probe-responses under anaesthesia.** (**A**) Representative example unit firing in response to the context-probe paradigm using "chimera" sequences. Top left: spectrogram of a sequence of echolocation pulse following the temporal pattern of the communication call used as context (called "fast echolocation"). Top right: spectrogram of a sequence of communication calls with the temporal pattern of the echolocation sequence used as echolocation context (called "slow communication"). Below: the spiking responses to the respective sounds (context) followed by a matching probe sound (top 20 trials) and by a mismatching probe sound (bottom 20 trials). Probe sounds correspond to a single echolocation pulse and a communication call, respectively. The grey square is aligned to the onset of the last call that correspond to the mismatching probe sound in both cases. For each context, the respective value of stimulus-specific suppression (s.s.s.) is indicated. (**B**) Context effect after fast echolocation context (left) and after slow communication (*n* = 26). (**C**) Top: Boxplots of s.s.s. per context. Significance level below the boxplot indicates differences against null distribution (one-sample Wilcoxon signed-rank test). Bottom: percentage of units per quadrant as showed in Fig 6A and 6B but for units recorded in response to stimuli showed in "A". (**D**) Discriminability index calculated 60 and 416 ms after the offset of the fast echolocation context (top) and slow communication (bottom) and after no context. (**E**) Same units' comparison (*n* = 14) between natural temporal pattern context effect (red) and switched temporal pattern context effect (purple). Left: context effect after echolocation context; right: after communication context. All significance levels were obtained using the paired test Wilcoxon signed-rank. Data supporting panels B-E can be found in the files "all_data.mat," "inj_data.mat," "chi_data.mat," "pt_data.mat," and "all_ce.mat" in "Ephys data" folder in https://gin.g-node.org/Luciana/anesthesia_bats. The underlying codes for each panel are in the same location.

sessions ($n = 14$; Fig 9E). Responses in the same neurons showed that increasing the call rate of the echolocation pulses did not modify significantly the context effects on any of the probe-evoked responses (Fig 9E left; Wilcoxon signed-rank test $p = 0.1$ for matching, $p = 0.08$ for mismatching). As observed in the previous data set using natural sequences (Fig 7A), the suppression driven by echolocation context was stimulus-specific. Interestingly, this was independent on the call rate of the echolocation pulses (Fig 9E left; Wilcoxon signed-rank test $p = 0.004$ for natural, $p = 0.005$ for artificial). On the other hand, a general reduction of the context effect was observed after the "slow communication" context compared to the natural communication treatment (Fig 9E right; Wilcoxon signed-rank test $p = 9.7 \times 10^{-4}$ for matching, $p = 0.02$ for mismatching). In this case, decreasing the call rate of the communication context did not lead to s.s.s. on the probe responses, as well as with the original sequences of communication.

## An updated model predicts anaesthetics effects on auditory cortex

Our experimental results showed that the neuronal evoked responses under anaesthesia were smaller in terms of spiking rate compared to those in wakefulness. Therefore, we assumed that the effects of KX on context-dependent processing cannot be explained by an antagonist action of ketamine on inhibitory neurons. In all of these models, the effect of ketamine predicted an enhancement of the spiking activity during the context sequences (Figs 4D and S4D). In addition, our results using chimera sequences suggest that the asymmetrical modulation by KX on the context effects depends on the carrier frequency of the sequences used as context, more than on their temporal patterns. KX might be then affecting neuronal properties in a frequency-specific manner (as hypothesised in Fig 8B), which explain why the nonspecific model effects were not able to predict our findings (Figs 2 and 3). Accordingly, we modified the awake model in the same manner described in Figs 2 and 3, but assuming that anaesthesia affects only one channel input. We ran several models systematically changing channel-specific parameters: either the firing rate of one input or the respective presynaptic adaptation (Table 1; models 1 to 4). In addition, we examined how an effect on postsynaptic neuronal adaptation modified the frequency-specific effects (Table 1; models 5 to 8). All of the models presented contradictory results with our data and could therefore not explain our findings.

Next, we tested combining both frequency-specific effects: input firing rate and presynaptic adaptation together, as showed in Fig 3. First, we modified the parameters associated with the LF input (S8 Fig). In these models, the echolocation context effects were not affected by anaesthesia (S8A Fig), which agrees with our electrophysiological data. However, the communication context effect on the mismatching probe responses did not increase (S8B Fig right) and neurons only decreased spiking activity during communication context (S8C Fig), both contradicting the experimental results. We reasoned that in order to increase context effect on mismatching probe responses, it is necessary to implement an increment of the postsynaptic adaptation (see Fig 2C). Therefore, we ran the same simulations (showed in S8A–S8C Fig), but with an enhancement of postsynaptic adaptation relative to the awake model (S8D–S8F Fig). Indeed, these new simulations showed an increment of communication context effect on mismatching probe-evoked responses (S8E Fig). However, echolocation also showed an increment on context effects (S8D Fig). This was due to the fact that postsynaptic adaptation depends on activity of the inputs once they converge in the cortical neurons, and, therefore, the effect cannot be input specific and it affects both contexts. In addition, this last model did not change the fact that anaesthesia modulates, in a stronger manner, the evoked response during communication than during echolocation (S8F Fig).

**Table 1. Qualitative comparison between multiple models of frequency-specific anaesthesia effect and data obtained in vivo.** In all models, anaesthesia effects were associated with either HF tuned inputs or LF tuned inputs, but never to both. Frequency-specific effects were implemented as a decrease of the cortical input firing rate ("input") or an increase of presynaptic adaptation ("presyn"), either one of them (1 to 8) or both (9 to 12). In addition, some of the models can include an increment of postsynaptic adaptation ("postsyn"), which is not frequency specific (5 to 8 and 11 to 12).

| model | input LF | input HF | presyn LF | presyn HF | postsyn | # effects | contradictory output with in vivo data |
|---|---|---|---|---|---|---|---|
| 1 | X | | | | | 1 | decrement of spike rate during communication and no change during echolocation. |
| 2 | | X | | | | 1 | decrement of echolocation context effect. |
| 3 | | | X | | | 1 | increment of stimulus-specific suppression after communication context. |
| 4 | | | | X | | 1 | increment of stimulus-specific suppression after echolocation context. |
| 5 | X | | | | X | 2 | increment of echolocation context effect. |
| 6 | | X | | | X | 2 | decrement of stimulus-specific suppression after echolocation context. |
| 7 | | | X | | X | 2 | increment of echolocation context effect. |
| 8 | | | | X | X | 2 | increment of echolocation context effect on mismatching probe-response. |
| 9 | X | | X | | | 2 | decrement of spike rate during communication and no change during echolocation. |
| 10 | | X | | X | | 2 | decrement of echolocation effect on mismatch probe-responses and negligible communication effect. |
| 11 | X | | X | | X | 3 | increment of echolocation effect on both probe-responses |
| 12 | | X | | X | X | 3 | None |

We then modelled the effects of anaesthesia by modifying the input firing rate and the presynaptic adaptation associated with the HF channel (Fig 10). The simulations showed that:

i. for a certain parameter space, echolocation context on matching probe responses remained unchanged relative to the awake model (Fig 10A), due to a compensation of the effects (explained in Fig 3);

ii. echolocation context on mismatching probe responses showed less suppression compared with the awake model, which is not observed in our data (Fig 10A right);

iii. communication context slightly increased the suppression on both probe responses (Fig 10B), due to the presence of HF components in communication sounds; and

iv. the effect on the spiking response during echolocation was stronger than communication (Fig 10C), which agrees with our data.

Interestingly, after combining the described channel-specific effects with an enhancement of postsynaptic adaptation, the outcomes reproduced the totality of our experimental data (Fig 10D–10G). The compensation region of parameters was still present (Fig 10D left), the echolocation effect on mismatching probe was compensated by the increment of suppression (Fig 10D right), and communication context suppression on both probes was accentuated (Fig 10E).

As described above, to reproduce the totality of our experimental data under anaesthesia, we had to change frequency channel-specific parameters in the model in a manner that these parameters compensated each other. To visualise this, we plotted the context effect and s.s.s. for a single combination of parameters within the region of compensation of effects and compared them with the results obtained from the awake model (Fig 10F and 10G). The results showed no changes in echolocation context effects between the awake and the anaesthesia model, with a significant difference between probe responses in both cases (Fig 10F left; Wilcoxon rank-sum test, $p = 1.5 \times 10^{-14}$ for echolocation awake and $p = 5.7 \times 10^{-15}$ for echolocation anaesthetized). On the other hand, an increment of suppression was observable on both probe-evoked responses after the communication context with the anaesthesia. As a consequence of this, the significant differences between probe responses disappeared in the

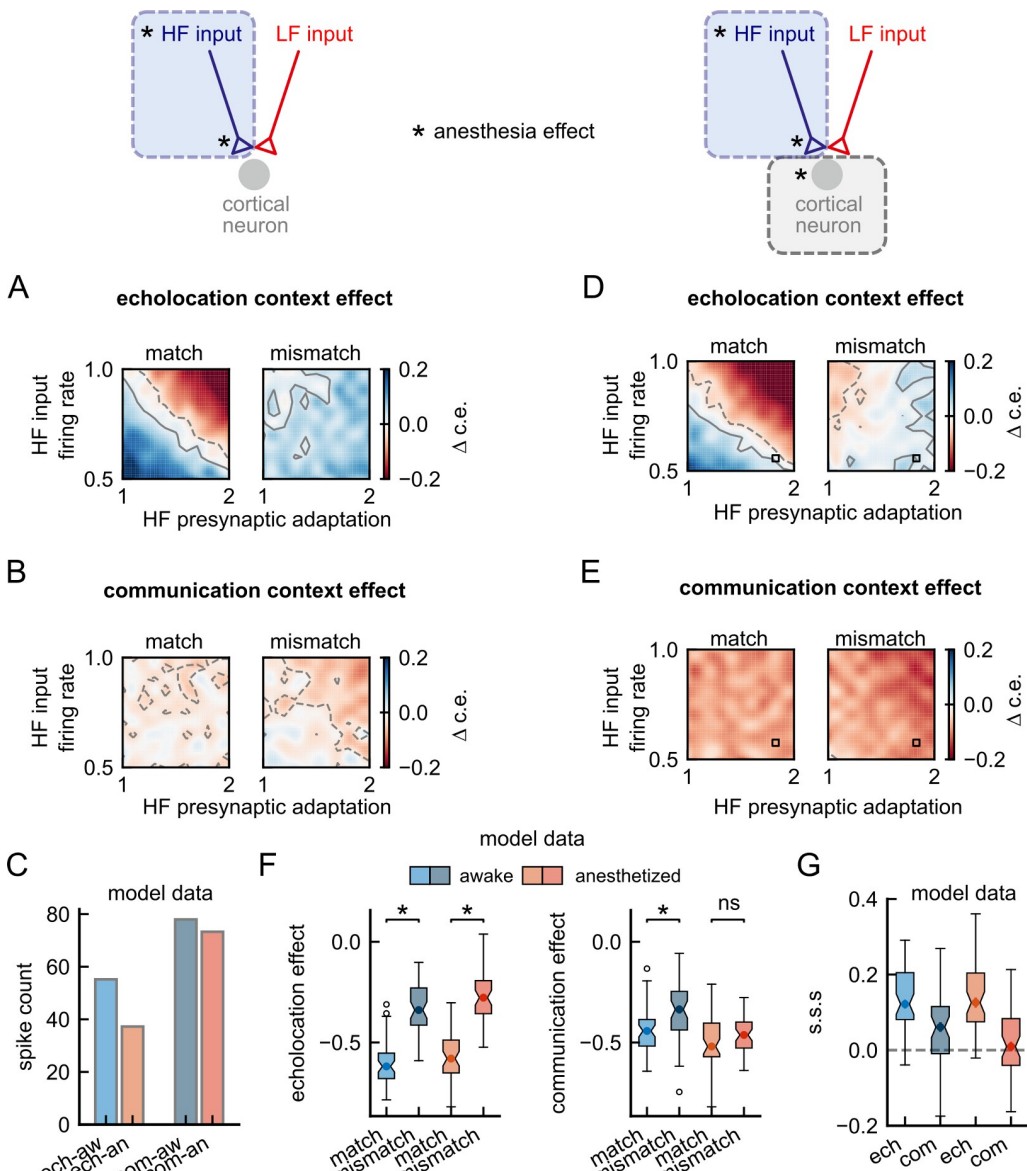

**Fig 10. Channel-specific anaesthesia effects on high-frequency cortical inputs can explain the experimental results.**
(**A-C**) Results obtained from an updated model in which anaesthesia affects only HF cortical inputs, decreasing input firing rate and increasing presynaptic adaptation. (**D-G**) Results obtained from a model that includes in addition to those described in "A-C", an increment of postsynaptic adaptation. (**A**) Difference between echolocation context effect when decreasing HF inputs firing rate and increasing HF presynaptic adaptation relative to the respective value obtained with the awake model. Contour lines in grey indicate the area with negligible effect size. Positive contours are indicated by solid lines and negative, by dashed lines. The anaesthesia effects were implemented as described in Fig 2. Note that red colours in the colormaps correspond to an increment of suppression, and blue, a decrement. (**B**) Same than "A", but difference between communication context effects. (**C**) Spike counts during the echolocation context and communication context; for the awake and anaesthetized models with HF input firing rate = 0.5 and HF presynaptic adaptation = 1.7. (**D, E**) Same than "A" and "B", respectively, but for a model that includes postsynaptic adaptation. (**F**) Left: context effect after echolocation context, in blue, for the awake model; in red, model under anaesthesia (parameters used are indicated with a square in D and E and correspond to HF input firing rate = 0.5 and HF presynaptic adaptation = 1.7). Right: context effect after communication context. (**G**) Stimulus-specific suppression (s.s.s.) per context for model awake and for model anaesthetized. The significance levels were obtained using the Wilcoxon rank-sum test. Data supporting this figure can be found in "coefC-HF_d-HF.npy" and "coefC-HF_d-HF_15-tau_th.npy" in "Model" folder in https://gin.g-node.org/Luciana/anesthesia_bats. The underlying codes are in "plots.py" in the same location.

anaesthesia model (Fig 10F right, $p = 1.0 \times 10^{-4}$ for communication awake and $p = 0.09$ for communication anaesthetized). Finally, the distribution of s.s.s. was very close to zero only after communication under anaesthesia (Fig 10G), as observed in our experimental data.

## Two excitatory conductances in pyramidal neurons better explain experimental data

To propose a model with a better biological correlate of our results, we built a slightly different cortical neuron model that includes two types of excitatory currents, opposed to just one. Here, the assumption is that these two excitatory currents are associated with distinct synaptic receptors, and ketamine affects only one of them (Fig 11A). In the established awake model with a single excitatory conductance, the parameters associated with synaptic adaptation depend on the tuning of the respective input. In this alternative model, these parameters were linked to different excitatory conductances. To illustrate the latter, we explored the adaptation of the synaptic receptors in response to constant stimulation. We ran a simulation of a cortical neuron receiving a regular spiking input at 25 Hz for 1.6 seconds. During the first spikes, at the beginning of the stimulation, the changes in conductance did not differ across the receptors. However, after 1.4 seconds of regular input, the amplitude of the conductance associated with each synapse differed (Fig 11B).

Next, we tested whether, in order to reproduce the experimental results, these two synaptic receptors ($R_1$ and $R_2$) need to be associated with one type of input or not (model A, Fig 11A or model B, Fig 11C). Considering the previous results (Table 1 and Fig 10), we assumed that anaesthesia affects the HF-input firing rate and the dynamic of the synaptic depression (i.e., presynaptic adaptation) associated with only one synaptic receptor ($R_1$). We modified both parameters simultaneously and calculated the difference in s.s.s. between the awake model and the subsequent anaesthetized models for both model configurations (Fig 11D). A compensation between the two effects arose in echolocation contexts in both types of models, which agrees with our experimental results. However, in communication contexts, a decrement in the s.s.s. under anaesthesia was observed only in model B. Thus, a model that reproduces the totality of our experiments and provides a plausible biological explanation of the effects of ketamine is a model in which pyramidal neurons possess (at least) two types of receptors associated with different input frequency tuning, and ketamine affects selectively one of them.

## Discussion

In this study, we show that known effects of anaesthesia on cortical neurons are not enough to explain the effects on context-dependent processing of vocalisations in the auditory cortex. In awake bats, both echolocation and communication acoustic contexts led to s.s.s. of lagging sounds. This phenomenon has been previously explained with a neuron model that includes convergence of narrowly tuned inputs and synaptic depression [23]. Here, we used the available model to predict effects of ketamine anaesthesia based on the known mechanisms of action of this drug. We called this step naïve modelling. None of the naïve model predictions were confirmed by actual experiments under anaesthesia. The in vivo experiments showed that only echolocation contexts drive s.s.s., indicating that anaesthesia has asymmetrical effects on context-dependent processing. The model of anaesthesia was updated in light of our experimental data. A model that satisfies all experimental results predicts that anaesthesia has a selective effect on different excitatory synaptic receptors in cortical neurons and affects particularly inputs tuned to HF sounds.

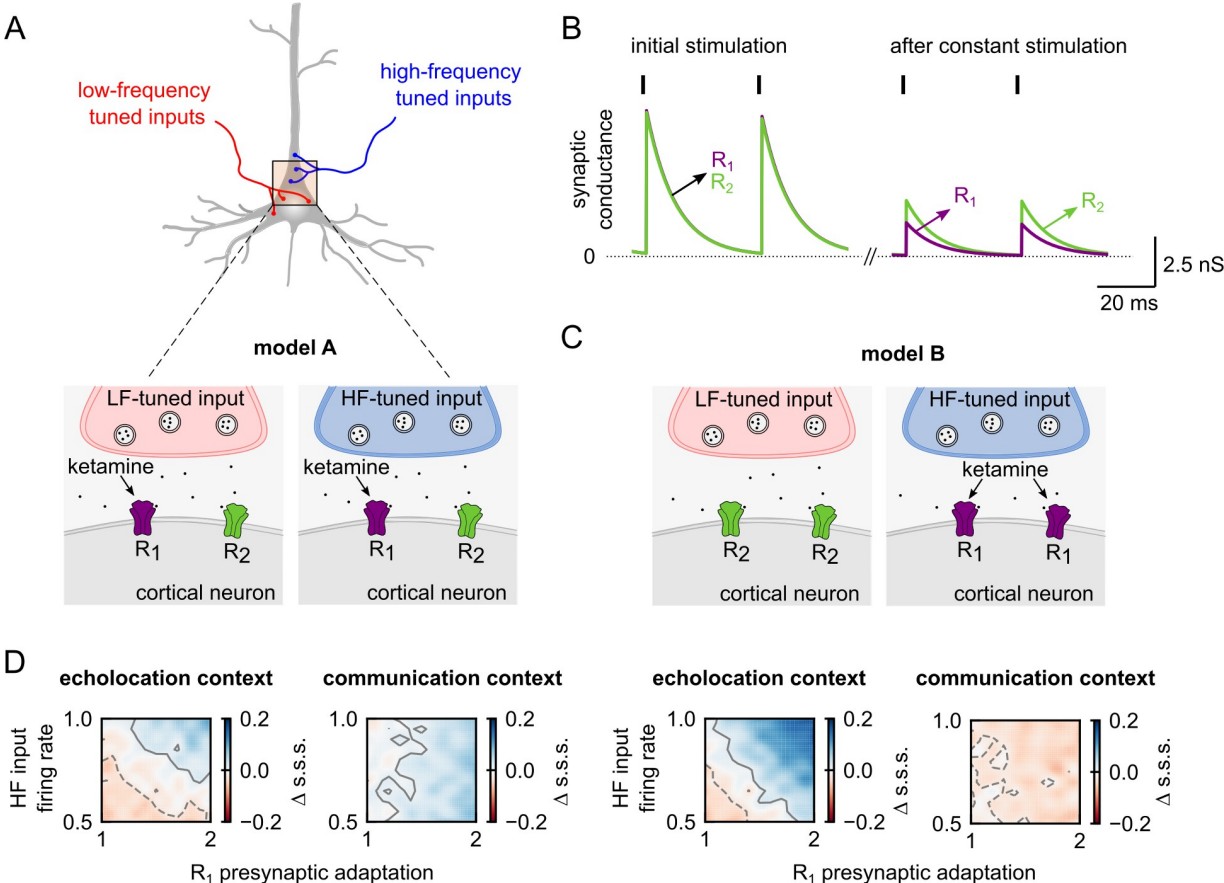

**Fig 11. A cortical neuron model with two distinct excitatory conductances provides plausible explanation to channel-specific effects of anaesthesia.** (**A**) Diagram of a pyramidal neuron receiving convergent inputs tuned to low- and high-frequency sounds. Bottom: sketch showing the presence of two receptors in the pyramidal neuron (model A). In this model, each cortical input activates receptors $R_1$ and $R_2$. (**B**) Changes in the synaptic conductance associated with each receptor, $R_1$ and $R_2$, in response to a regular spike train at 25 Hz. At the left, the two presynaptic spikes correspond to the third and fourth spikes of the spike train. At the right, two spikes after 1.4 seconds of periodic stimulation. Purple trace corresponds to receptor $R_1$ in which the adaptation rate is slower than in $R_2$ ($R_2$ = green trace). (**C**) Similar to A, but in this case, receptors $R_1$ and $R_2$ are associated with cortical inputs with different frequency tuning (model B). (**D**) Below the diagrams of model A and model B, the respective difference in stimulus-specific suppression indices (s.s.s.) between the awake model and different anaesthetized models. The anaesthesia effects were modelled by decreasing HF inputs firing rate and increasing presynaptic adaptation associated with receptor $R_1$. Left: after echolocation context; right: after communication context. Contour lines in grey indicate the area with negligible effect size. Positive contours are indicated by solid lines and negative with dashed lines. The anaesthesia effects were implemented as described in Fig 10. Data supporting panel D can be found in "modelA_coefC-HF-d_HF.npy," "modelA_coefC-HF-d_HF_15-tauth.npy," "modelB_coefC-HF-d_HF.npy," and "modelB_coefC-HF-d_HF_15-tauthv2.npy" in "Model" folder in https://gin.g-node.org/Luciana/anesthesia_bats. The underlying codes are in "plots.py" in the same location.

### Anaesthesia unbalances inputs in multifunctional neurons in the auditory cortex

Multifunctional neurons in the bat cortex process two types of vocalisations associated with different behaviours: echolocation and communication. We demonstrated that the effects of KX on these types of neurons cannot be generalised for both types of vocalisations. These asymmetrical effects between echolocation and communication neural processing can be explained by unbalanced synaptic inputs caused by anaesthesia. Multifunctional neurons integrate inputs across multiple frequency bands and their responses to natural sounds result from complex interactions between converging pathways. Other studies have also argued that anaesthesia unbalances synaptic inputs in the cortex [36–38]. Here, we demonstrated that the effects of

anaesthesia do not reduce uniformly the neurons responsiveness but interact in a complex manner to produce nontrivial effects on the spiking responses, which are ultimately measured as readout of cortical activity. Indeed, we showed that a compensation between several effects of KX may underlie seemingly unaffected neuronal responses in the frame of context-dependent processing (Fig 3). Particularly, we showed this might occur in the processing of vocalisations after echolocation pulses (Fig 10D). These results highlight the importance of using computational modelling approaches to understand mechanisms that underlie multifactor processes.

### Effects of ketamine on neuronal responses

**The activity of high-frequency tuned inputs to pyramidal neurons is selectively reduced.** Simulations under anaesthesia showed that a reduction in the firing rate of cortical inputs tuned to HF sounds is necessary to reproduce the data obtained in vivo. A frequency-specific decline of neuronal activity could arise from a number of sources, including both the central and the peripheral auditory system. For example, because ketamine selectively blocks NMDA receptors in central neurons, a variation of the effects of these receptors across the tonotopy could explain a frequency-specific effect. Such variations have been shown in terms of neuron excitability [39], amount of receptors [40], and synaptic properties [41]. A previous study in the same bat species used here showed that KX reduced inner ear sensitivity preferentially at high frequencies, presumably as a consequence of a metabolic influence on the cochlear amplifier caused by ketamine [42]. Similar mechanisms may explain age-related hearing loss in an energy-starved cochlear amplifier [43]. These possible changes in cochlear physiology under anaesthesia may affect central neuron responses, mainly to faint HF sounds, and their transmission to cortical regions. In agreement with the latter, here we showed that KX significantly reduces the cortical responses to the echolocation context, which consisted of a sequence of several biosonar pulses and their respective faint echoes.

**Anaesthesia increases nonspecific adaptation of cortical neurons.** To reproduce the anaesthetized in vivo data, we modified the input-unspecific adaptation of the awake neuron model by increasing the time constant of the postsynaptic adaptation process. Consequently, the anaesthetized model had slower dynamics of adaptation than the awake, resulting in a model with slower recovery of the suppressive effects after repeated stimulation during context sequences. Consistent with this, our experimental results showed that the recovery of the context effect was slower under anaesthesia (comparisons between gaps; Fig 7D and 7E). Previous studies have shown that, in general, anaesthetics increase the duration of the suppression by preceding stimulus in the auditory cortex [24–26,44].

**Excitatory synaptic receptors are affected asymmetrically by anaesthesia.** Although the decrement of the input activity may be inherited from subcortical areas, the predicted increase of presynaptic adaptation is likely a cortical process. Our model suggested that anaesthesia enhances synaptic depression, predominantly associated with the processing of HF sounds (Fig 10). Although the model with a single excitatory conductance reproduced all the experimental results (Fig 10), we decided to further explore models that provides a more biologically plausible explanation for frequency-specific synaptic effects of anaesthesia (Fig 11). Even though both models behave similarly and reproduce the experimental data equally well, they differ conceptually. The last model assumes the presence of two different synaptic receptors on cortical neurons and an asymmetrical effect of ketamine on them. The dynamic of these receptors varies only when there is considerable amount of synaptic depression, which only occurs during longer sounds, i.e., in acoustic contexts. The excitatory synaptic transmission in the cortex is mainly mediated by glutamate, which coactivates postsynaptic AMPA and NMDA receptors [45]. We hypothesise that the changes between the adaptations of the modelled

receptors can be associated with different ratios of NMDA and AMPA receptors. The latter is compatible with the antagonist action of ketamine on NMDA, but not on AMPA receptors. Consequently, synapses associated with higher ratios of NMDA would be more affected by ketamine than those with higher amount of AMPA receptors. To our knowledge, excitatory synaptic receptors with different depression dynamics in pyramidal neurons have not been reported in the cortex. We speculate that a heterogeneous distribution of synaptic receptors and/or heterogeneous innervation patterns in these multifunctional neurons may explain a differential gain in the effects observed under anaesthesia.

### Effects of anaesthesia on interneurons and their role in context-dependent processing

There is evidence that ketamine can selectively block glutamatergic receptors associated with inhibitory neurons causing disinhibition in the cortex [30]. Therefore, to understand the effects of KX, the incorporation of inhibition in the circuit modelled becomes crucial. In agreement with the disinhibition hypothesis, the naïve models including interneurons predicted an enhancement of cortical responses to sounds with anaesthesia (Figs 4D and S4D). Such increment in the spiking of pyramidal neurons was equivalent in the processing of calls in silence and after context. Consequently, no effective changes were observed in the s.s.s. in these models (Figs 4E and S4E). None of these predictions were confirmed with the actual experiments performed under anaesthesia. Although anaesthesia showed asymmetrical effects on the stimulus-evoked activity in our recordings, the effects were solely suppressive. This suggests that the effects of ketamine in context-dependent processing cannot be explained by its effect on inhibitory interneurons. This does not discard the possibility that inhibitory neurons contribute to the cortical phenomenon of context effect on acoustic responses in awake bats.

Several studies comparing data in awake and anaesthetized animals have reported response enhancements and no-effects of anaesthesia [46,47]. However, in the majority of the studies, suppression driven by ketamine seems to be the most frequent effect in the processing of natural sounds [12,48]. Interestingly, studies using artificial sounds in anaesthetized animals reported stronger activation of neurons in response to sounds than in awake conditions [27,49]. Contradictory effects of anaesthesia in the processing of sounds can be explained by differences in (i) the methods used to measure neural activity: population activity versus single neuron spiking; (ii) the area and the type of neurons recorded; (iii) the animal model; and (iv) the sound used to stimulate the neurons: i.e., artificial versus natural sounds.

Taken together, the results presented in this article indicate that, while useful, KX anaesthesia unbalances cortical inputs and have significant impact on neurons that integrates information across several frequency bands. We propose that the variability of effects in the processing of vocalisations found along the literature [11,12,44,47,50,51] is explained by the diversity and distribution of synaptic receptors and the afferents of the cortical neurons. Consequently, the effects on stimulus-evoked responses are difficult to predict without combining computational modelling and controlled in vivo experiments.

## Materials and methods

### Electrophysiological recordings

Recordings were conducted in 14 bats (10 males) taken from a breeding colony at the Institute for Cell Biology and Neuroscience at the Goethe University in Frankfurt am Main, Germany. Surgery and electrophysiological recordings were performed following the same procedures and experimental setup described elsewhere [23]. All experiments were conducted in

accordance with the Declaration of Helsinki and local regulations in the state of Hessen (Experimental permit #FU1126, Regierungspräsidium Darmstadt).

To perform the surgery, bats were anaesthetized with a mixture of ketamine (100 mg/ml Ketavet; Pfizer) and xylazine hydrochloride (23.32 mg/ml Rompun; Bayer). Under deep anaesthesia, the dorsal surface of the skull was exposed. The underlying muscles were retracted with an incision along the midline. A custom-made metal rod was glued to the skull using dental cement to fix the head during electrophysiological recordings. After the surgery, the animals recovered for at least 2 days before participating in the experiments.

On the first day of recordings, a craniotomy was performed using a scalpel blade on the left hemisphere in the position corresponding to the auditory region. Particularly, the caudoventral region of the auditory cortex was exposed, spanning primary and secondary auditory cortices (AI and AII, respectively), the dorsoposterior field (DP), and HF fields. The location of these areas was made following patterns of blood vessels and the position of the pseudocentral sulcus [34,52]. Bats were head-fixed and positioned in a custom-made holder over a warming pad whose temperature was set to 27˚C. A dose of 0.03 ml of the KX (same concentration used in the surgery) per 20 g of body mass was injected subcutaneously at the beginning of the recording session (actual dose: 7.5 mg/Kg of ketamine and 16.5 mg/Kg of xylazine). In addition, local anaesthesia (Ropivacaine 1%, AstraZeneca GmbH) was administered topically. Each recording session lasted a maximum of 4 hours. Experiments were made with at least 1-day recovery time in between. In all bats, recordings were performed over a maximum of 14 days.

All experiments were performed in an electrically isolated and sound-proofed chamber. For neural recordings, carbon-tip electrodes (impedance ranged from 0.4 to 1.2 MΩ) were attached to a preamplifier of DAGAN EX4-400 Quad Differential Amplifier system (gain = 50, filter low cut = 0.03 Hz, high cut = 3 kHz). A/D conversion was achieved using a sound card (RME ADI-2 Pro, SR = 192 kHz). Electrodes were driven into the cortex with the aid of a Piezo manipulator (PM 10/1; Science Products GmbH).

Single-unit auditory responses were located at depths of 266 μm +− 60 (mean +− SD), using a broadband search stimulus (downward frequency modulated communication sound of the same bat species) that triggered activity in both LF and HF tuned neurons of layers III to IV in the auditory cortex.

## Acoustic stimulation

We used bat vocalisations to trigger neural activity during the neuronal recordings. The natural sounds were obtained from the same species in previous studies from our lab [53,54]. Acoustic signals were digital-to-analog converted with an RME ADI.2 Pro Sound card and amplified by a custom-made amplifier. Sounds were then produced by a calibrated speaker (NeoCD 1.0 Ribbon Tweeter; Fountek Electronics, China), which was placed 15 cm in front of the bat's right ear. The speaker's calibration curve was calculated with a Brüel & Kjaer microphone (model 4135). In addition to the natural sounds, we presented pure tones to the animals to determine the iso-level frequency tuning of each neuron recorded. The pure tones had a duration of 20 ms with 0.5 ms rise/fall time. They were presented randomly in the range of frequencies from 10 to 90 kHz (5 kHz steps, 20 trials) at a fixed sound pressure level of 80 dB SPL. The intertone interval was 500 ms.

We studied context-dependent auditory responses using the paradigm described in our previous work [23]. Two types of acoustic contexts were presented before probe sounds. The contexts were sequences of echolocation pulses (S1A Fig) and sequences of distress calls, called "communication" throughout the text (S1B Fig). The probe sounds were a single echolocation pulse and an individual distress syllable (S5C Fig). The time interval between the context offset

and probe onset was either 60 ms or 416 ms. Therefore, a total of 8 context-stimuli (2 contexts × 2 probes × 2 gaps) were randomly presented and repeated 20 times to the bats during the electrophysiological recordings. In addition, each probe was presented in the absence of context, i.e., after 3.5 seconds of silence and repeated 20 times as well.

To study how the temporal modulation of context sequences affected our results, we modified the temporal pattern of the natural sequences used as context. These artificial "chimera" sounds corresponded to sequences of echolocation calls following the temporal pattern observed in the natural distress sequence and vice versa. To sweep the temporal pattern of the contexts, we first determined the onset of each call within the sequences as the time at which the envelope of the acoustic signal, measured using the secant method, surpassed a threshold of 70 times the standard deviation above the mean of a silent period of the signal. The silent interval was manually determined, and it corresponded to 260 ms in the communication sequence and 54 ms in the echolocation sequence. The secant method was performed with a MATLAB function called *env_secant* [55] using a window length of 200 for communication and 50 points for echolocation. To avoid false call-onset detections, we set a refractory period of detection of 5 ms and 0.8 ms, respectively. Secondly, each call within the natural sequences was replaced by the calls of the other sequence, keeping the same duration and root-mean-square (RMS) as the original. The RMS was adjusted to match each call-to-call interval. Finally, a calibration was performed with a microphone (model 4135; Brüel & Kjaer) in order to ensure that the chimera sequences had the same RMS than the original natural sounds.

## Paired datasets

In order to directly compare the effects observed between different preparations, we performed separate experiments in which the same neurons were recorded in the two different conditions: (i) in response to natural and constructed contexts; and (ii) awake and under anaesthesia. Once we found an auditory neuron, the natural and the constructed contexts were randomly presented after the probes using a gap of 60 ms. We excluded stimuli with longer gaps of 416 ms under this condition since the goal was only to corroborate the previous results, obtained across different sets of neurons. Next, we directly tested the effect of KX on the neuronal activity. For this purpose, before the recording sessions, we inserted and fixed a cannula subcutaneously in the lower back of the animal. We performed the neuronal recordings in the awake animal, and once the recording was finished, we injected the KX mixture (same concentration than the one used in anaesthetized recordings) via the cannula without opening the recording chamber. To monitor the state of the animal, we placed a ceramic piezo vibration sensor (DollaTek) between the bat abdomen and our custom-made holder. This sensor allowed us to record chest movements, as a measure of the respiration rate of the bat. The recordings under anaesthesia were performed only when the respiration rate was steady, lower in amplitude respective to the awake condition and in the absence of animal movements (S7 Fig top). We were able to record 10 neurons in a total of 5 animals.

## Data analysis

All the recording analyses, including spike sorting, were made using custom-written Matlab scripts (R2018b; MathWorks). The raw signal was filtered between 300 Hz and 3 kHz using a bandpass Butterworth filter (third order). To extract spikes from the filtered signal, we detected negative peaks that were at least three standard deviations above the recording baseline; signal in the times spanning 1 ms before the peak and 2 ms after were considered as one spike. The spike waveforms were sorted using an automatic clustering algorithm,

"KlustaKwik," which uses results from PCA analyses to create spike clusters [56]. For each recording, we considered only the spike cluster with the highest number of spikes.

**Electrophysiological classification of units.** Neurons were considered as auditory if the number of spikes was above the 95% confidence level calculated for spontaneous firing for the same unit (calculated along 200 ms before the start of each trial) for at least 8 ms after the onset of any probe sound in isolation. All the 149 auditory neurons recorded in the current study were classified in terms of the shape of their iso-level frequency tuning curves and in terms of their responses to the probe sounds in isolation (echolocation pulse and distress syllable). The frequency tuning curves were either single-peaked or multipeaked. In order to classify them, the spike count within a time window of 20 ms after the onset of each pure tone was averaged across trials and rescaled by minimum/maximum normalisation between the tested frequencies. Single-peaked units were defined as units whose normalised response was higher than or equal to 0.6 only to one half of frequencies measured: either lower than 50 kHz (LF tuned) or higher than 50 kHz (HF tuned). On the contrary, multipeaked neurons were those in which the normalised response exceeded 0.6 in both frequency bands ($<50$ and $>= 50$ kHz). For units that were responsive to both probe sounds in isolation, we categorised them by their preference. Neuronal preferences were determined by comparing the distributions of spikes count during 50 ms after the probes onset, using a nonparametric effect size metric: Cliff's delta. The present study focused exclusively on those units that presented a negligible or small effect size (abs(Cliff's delta)$< = 0.3$), called "equally responsive" units. These are units that responded equally well to communication and echolocation sounds presented in isolation. This type of units was previously characterised by us, using the same criterion mentioned above [23]. The rest of the neurons presented either preference for one of the probe sounds (abs(Cliff's delta)$>0.3$) or unique response to one sound and were excluded from the current study. The number of units per classification, namely "equally responsive," "preference to communication," "preference to echolocation," "echolocation only," and "communication only," is plotted in S9 Fig. The classification was grouped by experimental condition: awake, anaesthetized in response to natural sounds, anaesthetized in response to artificial-natural sounds, anaesthetized in response to both, natural and artificial-natural sounds, and before and after anaesthesia injection in response to natural sounds.

**Quantification of context-dependent processing.** The effects of the leading acoustic context on probe-triggered responses were quantified by the indices "context effect," "stimulus-specific suppression," and "discriminability index."

Context effect quantifies the effect of context (whether echolocation or communication) on the response to a lagging probe sound. This index was calculated as follows: $(R_{C,p} - R_{NC,p})/(R_{C,p} + R_{NC,p})$, where $R_{C,p}$ and $R_{NC,p}$ corresponds to the number of spikes during 50 ms after the onset of the probe $p$ followed by the context $C$ and followed by 3.5 seconds of silence (no context, $NC$), respectively.

Stimulus-specific suppression index was defined as: $(E_{C,m} - E_{C,mm})/2$, where $E_{C,m}$ is the context effect for context $C$ and probe $m$, in which $m$ *matches* with context $C$ (i.e., echolocation context and echolocation probe or communication context and communication probe) and $E_{C,mm}$ for context $C$ and probe $mm$, which *mismatches* with the context $C$ (echolocation context and communication probe or communication context and echolocation probe). The index gives values between $-1$ and 1. Positive values indicate stronger suppressive effect of the context on matching probe responses relative to mismatching probe responses.

Discriminability index corresponds to a nonparametric effect size measure that quantifies the difference between the responses to both probes across trials. It was calculated for each context using the Cliff's delta statistic ($d$) between the spike counts during 50 ms after each

trial presentation of echolocation ($x_i$) and communication ($x_j$) probe sounds, as follows:

$$d = \frac{\sum_{i,j}[x_i > x_j] - [x_i < x_j]}{mn}$$

where $m$ and $n$ are the total number of trials for each probe presentation, and $[\cdot]$ is 1 when the contents are true and 0 when false. According to this, negative Cliff's delta values indicate higher responses to the communication probe than to the echolocation probe. Positive Cliff's delta values indicate the opposite.

## Computational modelling of anaesthesia effects

We modified an existing model [23] implemented as an integrate-and-fire neuron model. The simulations were run using the Python package Brian2, version 2.3 [57]. The model consists of a neuron with subthreshold dynamics that integrates spiking inputs from two narrow frequency channels. The output reproduces the behaviour of the neurons observed in our experiments in awake animals. The synapses are excitatory and exhibit activity-dependent depression. In addition, the neuron presents an adaptive firing threshold that allows it to adapt to inputs that arrives close in time. To study the effects of anaesthesia, we modified three parameters: average input rate $v$, synaptic decrement input $\Delta_{j,s}$ and adaptive threshold time constant $\tau_{th}$. The variation of the parameters was systematically tested by multiplying the original values (awake model) by a vector of equidistant numbers. Therefore, a factor of 1.0 represented the results obtained in the awake model. Throughout the study, the vectors were consistent: for input rate, the vector went from 1.0 to 0.5 in steps of 0.05; in the case of synaptic decrement input, from 1.0 to 2.0 in steps of 0.1 and for varying the adaptive threshold time constant, the vector used went from 1.0 to 2.5 in steps of 0.1.

In the previous model, the inputs to the neuron were modelled by an inhomogeneous Poisson process whose rate is proportional to the envelope of the natural sounds used to stimulate the neurons. The amplitude of the envelope is pondered by the average input rate $v$ as well as by a factor $k_{j,sound}$ that represent the responsivity of input $j$ to the sound, either echolocation or communication. In order to test the effect of the input rate in the output of the model, we varied the value of $k_{j,ech}$, $k_{j,com}$ either identically for both inputs (Figs 2 and 3) or asymmetrically, only for one input (LF input: S8 Fig and HF input: Fig 10). Regarding the increment of presynaptic adaptation by means of the parameter synaptic decrement input $\Delta_{j,s}$, the variations were either implemented on both inputs (Figs 2, 3 and S2B) or only for one input (LF input: S8 Fig and HF input: Fig 10). Finally, modifications of the adaptive threshold cannot be input specific since the parameter acts on the postsynaptic neuron. We systematically tested different values using the vector mentioned above (Figs 2 and S2), and also, the effect of an increase by a factor of 1.5 on the combination of input rates and presynaptic adaptation effects (Figs 10 and S8).

The simulations consisted of 50 neurons, with 20 trials each, in response to the context-probe paradigm. The spiking of the neuron model was analysed exactly as it was done with the electrophysiological data. The spike counts to calculate context effects and responses during contexts were calculated in the same time windows used with the in vivo data. Along the study, several anaesthetized models were compared against the original awake model. To quantify the size of the effect on the outputs of the anaesthetized models versus the awake model, we used Cliff's delta statistics. The effect size was interpreted as negligible, small, medium or large defined by the limits 0.147, 0.333, and 0.474, respectively [58].

**Cortical circuit with inhibitory neurons.** We implemented two alternative models that include inhibitory interneurons in the cortical circuit. In the first model, interneurons are reciprocally connected with the pyramidal excitatory neurons [59]. In the second,

interneurons and pyramidal neurons share common inputs (31). In both models, an inhibitory synaptic conductance was added to the stablished neuron model that lacked interneurons, resulting in a new pyramidal neuron model that evolves according to the equation:

$$\frac{dV_m}{dt} = \frac{g_L(E_L - V_m) + g_e(E_e - V_m) + g_i(E_i - V_m)}{C_m} + \sigma\sqrt{2/\tau_\sigma}x_i$$

where $g_L$ is the leak conductance, $E_L$, the resting potential, and $C_m$, the membrane capacitance. The excitatory current is defined by the reversal potential, $E_e$, and the dynamic of the conductance, $g_e$. As for excitatory synapses, the temporal course of the inhibitory conductance, $g_i$, was modelled with an single exponential decay. Although, the reversal potential of the inhibitory synapse, $E_i$, was set to −75 mV with a time constant of 5 ms. The rest of the parameters were the same than those used in the models without interneurons.

To model the interneurons, we implemented a model that consists on leak, excitatory, and stochastic currents, similar to those used to model the excitatory pyramidal neurons. The parameters that define the leak and the excitatory currents of the inhibitory neurons were the same than those set for the pyramidal neuron model (Table 2). It is known that the spontaneous firing rate of interneurons is 2-fold times higher than pyramidal neurons [60]. To reproduce this, we increased the variance $\sigma$ of the Ornstein–Uhlenbeck process that specifies the stochastic current, implemented as shown in the second term of the previous equation. The increment of this parameter doubled the spontaneous activity of this neuron model compared to the excitatory neuron model.

The inclusion of inhibition in the cortical neuron model reduced the firing rate of our stablished pyramidal neuron model. In order to reestablish the magnitude of the cortical responses, we adjusted the synaptic weights of the inputs to pyramidal neurons and

**Table 2. Parameters used in models that include inhibitory neurons.**

| Parameters | Values |
|---|---|
| *Inhibitory neuron model* | |
| $C_m$, membrane capacitance | 100 pF |
| $E_L$, leak reversal potential | −55 mV |
| $g_L$, leak conductance | 5 nS |
| $V_{th}$, firing threshold potential | −50 mV |
| $V_r$, reset potential | −55 mV |
| $\sigma$, sigma noise | 3 mV |
| $\tau_\sigma$, time constant noise | 10 ms |
| *Excitatory synapses on interneurons* | |
| $E_i$, excitatory reversal potential | 0 mV |
| $\tau_e$, excitatory time constant | 10 ms |
| *Inhibitory synapses on pyramidal neurons* | |
| $E_i$, inhibitory reversal potential | −75 mV |
| $\tau_e$, inhibitory time constant | 5 ms |
| $w_{i,e}$, synaptic weight | 2 nS |
| *Recurrent circuit model* | |
| $w_{e,e}$, synaptic weight input to pyramidal | 12 nS |
| $w_{e,i}$, synaptic weight input to interneurons | 2 nS |
| *Common input model* | |
| $w_{e,e}$, synaptic weight input to pyramidal | 12 nS |
| $w_{e,i}$, synaptic weight input to interneurons | 8 nS |

interneurons. First, the synaptic weight of the inhibitory synapse was set arbitrary in 2 nS. To obtain an average of approximately 5 spikes/50 ms in response to the probe sounds in silence, as observed experimentally, we increased the synaptic weight of the inputs to pyramidal neurons from 8 nS to 12 nS. The synaptic weights of the excitatory inputs to interneurons vary across the two alternative models (recurrent circuit model and common input model). In both models, they were lower than the weight associated with the inputs to pyramidal neurons. Values are shown in Table 2.

To test the effects of anaesthesia on both alternative circuits with inhibitory neurons, we systematically reduced the synaptic weight associated with the inputs of pyramidal neurons and interneurons. The value of synaptic weights set for the awake model were multiplied by a vector of equidistant decreasing numbers. For the synapses in the pyramidal neurons, the vector went from 1.0 to 0.75 in steps of 0.025; in the case of the synapses in the interneurons, from 1.0 to 0.5 in steps of 0.05.

*Two distinct excitatory conductances in pyramidal neurons.* To propose a model with a better biological correlate of our results, we slightly modified the cortical neuron model that reproduced the totality of the experimental data. Instead of a single excitatory conductance with different parameters depending on the type of the input, we implemented two different excitatory conductances with different dynamics, resulting in the following equation:

$$\frac{dV_m}{dt} = \frac{g_L(E_L - V_m) + g_{e,1}(E_{e,1} - V_m) + g_{e,2}(E_{e,2} - V_m)}{C_m} + \sigma\sqrt{2/\tau_\sigma}x_i$$

The excitatory reversal potentials, $E_{e,1}$ and $E_{e,2}$, the respective time constants, $\tau_{e,1}$ and $\tau_{e,2}$, and the synaptic weights, $w_{e,e,1}$ and $w_{e,e,2}$, were not different across the two excitatory currents (Table 3). However, the parameters associated with the synaptic depression of these receptors were different. In the model with a single excitatory conductance, the parameters associated with synaptic adaptation depend on the type of inputs that arrive to the synapses, either LF tuned inputs or HF tuned inputs. In this alternative model, these parameters were associated with one of the two excitatory conductances. Briefly, the effective synaptic weight of these synapses changes in time according to the variable $X_s$, which multiplied by a fixed value of synaptic weight, set the effective weight of each synapse. Initially, $X_s$ starts at 1.0 and each

**Table 3. Parameters used in a pyramidal neuron model with two excitatory conductances.**

| Parameters | Values |
|---|---|
| *Excitatory receptor in pyramidal neurons, $R_1$* | |
| $E_{e,1}$, excitatory reversal potential | 0 mV |
| $\tau_{e,1}$, excitatory time constant | 10 ms |
| $W_{e,e,1}$, synaptic weight | 8 nS |
| $\Omega_{s,1}$, adaptation rate | 1.0/s |
| $\Delta_{s,1}$, synaptic decrement | 0.040 |
| *Excitatory receptor in pyramidal neurons, $R_2$* | |
| $E_{i,2}$, excitatory reversal potential | 0 mV |
| $\tau_{e,2}$, excitatory time constant | 10 ms |
| $W_{e,e,2}$, synaptic weight | 8 nS |
| $\Omega_{s,2}$, adaptation rate | 1.6/s |
| $\Delta_{s,2}$, synaptic decrement | 0.045 |

presynaptic spike arrival decreases the magnitude of $X_s$ in $\Delta_s$, which evolves in time following:

$$\frac{dX_s}{dt} = \Omega_s(1 - X_s)$$

where $\Omega_s$ represent the rate at which the synapse adapts. The values of the parameters $\Delta_s$ and $\Omega_s$ differ across conductances, similarly to those in the single excitatory conductance model (see Table 3).

## Supporting information

**S1 Fig.** Spectrogram of natural calls of *Carollia perspicillata* associated with echolocation and communication behaviours used to study context-dependent sensory processing of natural sounds. (**A**) A sequence of echolocation pulses followed by a probe sound that, in this case, is a single echolocation pulse. This example corresponds to a "matching" sound transition. (**B**) A distress call followed by a probe sound, i.e., the same echolocation probe used in A. This example corresponds to a "mismatching" sound transition going from communication (distress) to echolocation.
(EPS)

**S2 Fig.** (**A**) Difference between context effect (left) and stimulus-specific suppression (right) when decreasing inputs firing rate and increasing postsynaptic adaptation relative to the respective value used in the awake model. First row, after echolocation. Second row, after communication context. The anaesthesia effects were implemented as described in Fig 2. (**B**) Difference between context effect (left) and stimulus-specific suppression (right) when increasing presynaptic and postsynaptic adaptation relative to the respective value used in the awake model. First row, after echolocation. Second row, after communication context. The anaesthesia effects were implemented as described in Fig 2. Contour lines in grey indicate the area with negligible effect size. Positive contours are indicated by solid lines and negative, by dashed lines. Note that red colours in the colormaps of c.e. correspond to an increment of suppression, and blue, a decrement. Data supporting this figure can be found in "coefC_tau_th.npy" and "d_tau_th.npy" in "Model" folder in https://gin.g-node.org/Luciana/anesthesia_bats. The underlying codes are in "suppl_plots.py" in the same location.
(EPS)

**S3 Fig.** (**A**) Left: a diagram of a cortical circuit in which interneurons receive stronger inputs tuned to high-frequency sounds than inputs tuned to low-frequency sounds. Right: simulated responses to sounds of pyramidal neurons in the circuit at the left. The number of spikes was measured in a time window of 50 ms after the onset of each probe sound: echolocation (echo) and communication (com). Acoustic context was absent in the simulations. (**B**) Left: a diagram of a cortical circuit in which interneurons receive stronger inputs tuned to low-frequency sounds than inputs tuned to high-frequency sounds. Right: same than in A, but for the respective circuit illustrated at the left. (**C**) Left: a diagram of a cortical circuit in which interneurons receive balanced inputs tuned to high- and low-frequency sounds. Right: same than in A, but for the respective circuit illustrated at the left. The significance levels were obtained using the Wilcoxon rank-sum test.
(EPS)

**S4 Fig.** (**A**) Diagram of the circuit model. (**B**) Context effect on probe-responses following echolocation and communication contexts obtained from a model as illustrated in A. (**C**) Effects of ketamine on the synaptic weight of the excitatory inputs to the pyramidal neuron ($w_{e,e}$) and of the excitatory input to the interneuron ($w_{e,i}$) of the circuit model shown in A. (**D**)

Combined effect of the reduction of excitatory inputs to the pyramidal neuron and to the interneuron on the firing rate of the pyramidal neurons in response to the sequences of echolocation and communication calls used as contexts. (**E**) Same simulations than above but showing the variation in the stimulus-specific suppression index in relation to the awake model. Data supporting the panels B, D, and E can be found in "w1-w2v2.npy" in "Model" folder in https://gin.g-node.org/Luciana/anesthesia_bats. The underlying codes are in "suppl_plots.py" in the same location.
(EPS)

**S5 Fig.** (**A**) Coronal section showing the recording site in the bat auditory cortex (AC). The recording enter point was labelled by DiI (red mark) and the recording site calculated with the depth of the electrode (dashed line), indicated by the white arrow. Scale bar indicates 1 mm. The left lower diagram indicates the location of the section in the anterior–posterior axis of the bat brain. (**B**) Frequency tuning of a neurons recorded under anaesthesia from the high-frequency fields of bat AC. The firing rate was calculated during 50 ms after the tones' onset. Above: the respective iso-level frequency tuning curve recorded at 80 dB SPL. (**C**) Spectrograms of sounds used as probes: an echolocation pulse (left) and a distress syllable (right); below: the respective neuronal response of a representative multipeaked neuron that responded equally well to both sounds categories. (**D**) Number of units classified according to responses to echolocation and communication probes recorded in awake (left) and anaesthetized bats (right). Categories were defined considering the number of spikes evoked during 50 ms from the sound onset ["ech = com": abs(Cliff's delta) $< = 0.3$; "ech$<$com" or "com$<$ech": abs(Cliff's delta) $> 0.3$]. Colour code indicate the shape of the iso-level frequency tuning curve obtained at 80 dB SPL per natural sound category. Multi., multipeaked frequency tuning; LF, low-frequency tuned units; HF, high-frequency tuned units. (**E**) Percentage of units per preparation: awake and anaesthetized that corresponded to each frequency-tuning shape indicated in "d." (**F**) Average iso-level frequency tuning curve for units that were classified as low-frequency tuned (LF) and as multipeaked (multi.). Data supporting the panels B-F can be found in the files "12p1.mat," "inj_psth.mat," "class_FT-aw.mat," "class_FT-int.mat," "class_FT-chi.mat," "class_FT-pt.mat," "class_FT-pair.mat," "all_data.mat," "chi_data.mat," "pt_data.mat," and "pair_data.mat" in "Ephys data" folder in https://gin.g-node.org/Luciana/anesthesia_bats. The underlying codes for each panel are in the same location.
(EPS)

**S6 Fig.** Normalised and averaged firing rate of units that corresponded to the most representative categories in Fig 5B (quadrant I for awake and quadrant II for anaesthetized) aligned by probe onset. Blue background corresponds to echolocation probe and red, to communication. Left: for awake preparation; right: for anaesthetized preparation. The responses to the probes correspond to those that were preceded by silence.
(EPS)

**S7 Fig.** Seven units were recorded during the context-probe paradigm before and after anaesthesia injection. The anaesthesia was injected by using a cannula previously fixed subcutaneously to the animal. The respiration rate of the animal was monitored during the recordings to ensure the success of the anaesthesia injection. Top: black trace shows the piezo signal that monitor the abdominal movements of the bat together with the triggers to natural sounds playbacks. (**A**) Context effect on matching and mismatching probe-responses for the same neurons awake and under anaesthesia, after echolocation (left) and communication (right). The comparison before and after were not significant; the Cliff's delta value is indicated on top of the pairs. Grey lines joint the same units. (**B**) Spike counts for 1 second of silence, during

the echolocation context and communication context; for the same preparation awake and anaesthetized. The significance levels were obtained using a paired test (Wilcoxon signed-rank). Data supporting this figure can be found in the files "pair_data.mat" and "sp_context_-pair.mat" in "Ephys data" folder in https://gin.g-node.org/Luciana/anesthesia_bats. The underlying codes for each panel are in the same location.
(EPS)

**S8 Fig.** (**A**-**C**) Results obtained from a model in which anaesthesia affects only low-frequency (LF) cortical inputs, as decrement of input firing rate and increment of presynaptic adaptation. (**D**-**F**) Results obtained from a model that includes in addition to those described in "A", an increment in postsynaptic adaptation. (**A**) Difference between echolocation context effect when decreasing LF inputs firing rate and increasing LF presynaptic adaptation relative to the respective value obtained with the awake model. Contour lines in grey indicate the area with negligible effect size. Positive contours are indicated by solid lines and negative, by dashed lines. The anaesthesia effects were implemented as described in Fig 2. Note that red colours in the colormaps of c.e. correspond to an increment of suppression, and blue, a decrement. (**B**) Same than "A", but difference between communication context effects. (**C**) Spike counts during the echolocation context and communication context; for awake and anaesthetized models with LF input firing rate = 0.5 and LF presynaptic adaptation = 1.7. (**D**-**F**). Same than "A", "B", and "C", respectively, but for a model that include increment of postsynaptic adaptation as an effect of anaesthesia. Data supporting this figure can be found in "coefC-LF_d-LF.npy" and "coefC-LF_d-LF_15-tau_th.npy" in "Model" folder in https://gin.g-node.org/Luciana/anesthesia_bats. The underlying codes are in "suppl_plots.py" in the same location.
(EPS)

**S9 Fig.** Total number of units recorded for the current study. Note that the analysis presented in this paper only includes those units that were classified as "ech = com," highlighted in black. Natural sounds correspond to the sequences of echolocation and communication showed in S1 Fig. Chimeras sounds correspond to sequences of echolocation pulses and distress syllables but after switching the temporal patterns of the original sequences. The last two plots at the right correspond to "paired" data sets. Experiments in which two conditions were tested on the same neurons; either neuronal responses to chimeras sounds versus natural sounds, or neuronal responses in awake and anaesthetized animals. Data supporting this figure can be found in the files "all_data.mat," "chi_data.mat," "pt_data.mat," and "pair_data.mat" in "Ephys data" folder in https://gin.g-node.org/Luciana/anesthesia_bats. The underlying code is in the same location.
(EPS)

## Author Contributions

**Conceptualization:** Luciana López-Jury, Francisco García-Rosales, Eugenia González-Palomares, Julio C. Hechavarria.

**Data curation:** Luciana López-Jury.

**Formal analysis:** Luciana López-Jury, Michael Pasek.

**Funding acquisition:** Julio C. Hechavarria.

**Investigation:** Luciana López-Jury, Julio C. Hechavarria.

**Methodology:** Luciana López-Jury.

**Supervision:** Julio C. Hechavarria.

**Validation:** Luciana López-Jury.

**Visualization:** Luciana López-Jury, Michael Pasek.

**Writing – original draft:** Luciana López-Jury.

**Writing – review & editing:** Francisco García-Rosales, Eugenia González-Palomares, Johannes Wetekam, Michael Pasek, Julio C. Hechavarria.

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
