## [Editor Report · Decision Letter 0]

19 Jul 2022

Dear Dr Lopez Jury, 

Thank you for submitting your manuscript entitled "A neuron model with unbalanced synaptic weights explains asymmetric effects of ketamine in auditory cortex" for consideration as a Research Article by PLOS Biology.

Your manuscript has now been evaluated by the PLOS Biology editorial staff, as well as by an academic editor with relevant expertise, and I am writing to let you know that we would like to send your submission out for external peer review.

Once your full submission is complete, your paper will undergo a series of checks in preparation for peer review. After your manuscript has passed the checks it will be sent out for review. To provide the metadata for your submission, please Login to Editorial Manager (https://www.editorialmanager.com/pbiology) within two working days, i.e. by Jul 21 2022 11:59PM.

Kind regards,

Kris

Kris Dickson, Ph.D. (she/her)

Neurosciences Senior Editor/Section Manager

PLOS Biology

kdickson@plos.org

---

## [Decision Letter · Decision Letter 1]

26 Aug 2022

Dear Dr Lopez Jury,

Thank you for your patience while your manuscript "A neuron model with unbalanced synaptic weights explains asymmetric effects of ketamine in auditory cortex" was peer-reviewed at PLOS Biology. It has now been evaluated by the PLOS Biology editors, an Academic Editor with relevant expertise, and by several independent reviewers. 

In light of the reviews, which you will find at the end of this email, we would like to invite you to revise the work to thoroughly address the reviewers' reports. In particular, we ask that you 1) expand the complexity of the computational model to bolster your conclusions and provide further insights into the biological effects of ketamine as requested by Reviewer 2 and 2) given our broad biology readership, we also ask that you provide additional clarifications of your findings and their biological implications as noted in Reviewer 1's comments.

Given the extent of revision needed, we cannot make a decision about publication until we have seen the revised manuscript and your response to the reviewers' comments. Your revised manuscript is likely to be sent for further evaluation by all or a subset of the reviewers.

**IMPORTANT - SUBMITTING YOUR REVISION**

*Re-submission Checklist*

*Published Peer Review*

*PLOS Data Policy*

*Blot and Gel Data Policy*

Sincerely,

Kris

Kris Dickson, Ph.D. (she/her)

Neurosciences Senior Editor/Section Manager

PLOS Biology

kdickson@plos.org

REVIEWS:

Reviewer #1: In "A neuronal model with unbalanced synaptic weights explains asymmetric effects of ketamine in the auditory cortex", Lopez-Jury and colleagues use a combination of computational modeling and in vivo electrophysiology to understand the impact of ketamine/xylazine (KX) anaesthesia on auditory cortical neurons responses to various contexts. The two main contexts that are examined are an echolocation/navigation context and a social (distress call) context. The authors varied several parameters in a simplified cortical model - the degree of adaptation of synaptic inputs, the degree of pre-synaptic or the degree of post-synaptic depression induced by KX and examined how varying these parameters affected model responses to contextual stimuli. They then used hypotheses generated by these data to design in vivo experiments to test these hypotheses, including several experiments involving testing the same neurons pre- vs. during- KX anesthesia. They also designed novel experiments using chimeric stimuli to separate temporal from frequency effects, and ultimately concluded that KX effects are dominated by its impact on frequency. 

In general, I think this manuscript represents an excellent example of cross-fertilization of computational modeling with electrophysiology experiments. It is not clear to me that the results will have a general impact on our understanding of neural processing of context, or simply in how KX affects neural networks. The study is complicated and I had a hard time following the manuscript. I think it may help to make some small organizational changes in the manuscript. For example, it seems odd to begin the paper with PubMed data on the importance of KX. These types of data are typically found in grant proposals or lectures, not papers. I think it'd be better to have a figure 1 that really motivates the paper. What phenomenon are the authors attempting to understand? Clearly it is the effect of context on acoustic responses. I think providing the reader with a clearer example of the physiological phenomenon will help us to understand why the subsequent manipulations were made. The rest of my comments are minor:

1. "Navigation" and "echolocation" appear to be used interchangeably. Would suggest using just one.

2. The phrase "associated to" is used often. It should be "associated with"

3. Line 179, place a comma after "regions"

4. Line 258, should be "evoked"

5. Line 532, please provide actual dose (in mg or mg/kg) of ket and xylazine here

6. Fig 5A legend - spelling "anestheitzed"

7. In general for the figures, whenever rasters are shown, it would be helpful to have a diagram of the stimulus shown above or below the raster

8. Also, in general for the figures, it would be useful if somewhere on the figure it is stated if the data are real neuronal data or model data

Reviewer #2 (Jordan David Chambers): In this manuscript, the authors investigated how the auditory cortex responds to vocalizations under ketamine induced anaesthesia and in wakefulness. They used a combination of electrophysiological recordings and mathematical modelling. The electrophysiological recordings showed ketamine effects on context-dependent processing varied with the frequency composition of the sounds used as context. That is, ketamine effected communication (low frequency) context and not echolocation (high frequency) context. The mathematical modelling explored parameters relating to cortical input firing rate, presynaptic adaptation, and postsynaptic adaptation. The model only reproduced all electrophysiological recordings when ketamine affected only high-frequency cortical inputs and both presynaptic and postsynaptic adaptation.

Overall, the manuscript is well written. The methodology is sufficient to reproduce the experiments. The electrophysiological data and analysis fully support the author's claims. The computational model presented supports the author's claims. However, the computational model is relatively simple, which may or may not be misleading. The findings presented are novel and will be of interest to researchers in the field of auditory processing and more generally to sensory processing. The findings relating to ketamine and anaesthesia will be of interest to all neurosciences.

Major concerns:

1. Ketamine mainly affects glutamatergic transmission inhibiting NMDA receptors, but the computational model does not contain inhibitory neurons. The authors recognise this and discuss the limitations of their model but trying to model the affects of ketamine without inhibitory neurons will significantly reduce the impact of the computational model.

2. The mathematical model explores parameter spaces relating to presynaptic and postsynaptic adaptation. This adaptation is modelled by increasing the threshold, which then decays with a time constant. This sort of adaptation can easily be applied to any ion channels in the model, but this model only contains a single excitatory conductance (in addition to a leak conductance and a stochastic conductance). By including more conductances and exploring the adaptation for each conductance, the model could easily provide an explanation as to why ketamine is affecting high frequency presynaptic adaption and not low frequency presynaptic adaption (e.g. ketamine is affecting cells that contain a particular ion channel). At the moment, the computational model requires a different affect on low frequency input and presynaptic adaptation compared to the high frequency input and presynaptic adaptation. This is a not a model providing insights into how ketamine has different affects on communication context compared to echolocation context, but simply a model that separates the low and high frequencies to replicate the experimental observations. The authors do discuss the evidence for a peripheral affect in the cochlear, but if this is the only mechanism, then the findings are of less importance to the sensory processing and general neuroscience.

Minor comments:

1. Line 266-267 The sentence "The same gap was used in the stimuli used in…" is confusing.

2. Line 272: "Despite of" should just be despite?

3. Throughout the text adaption and adaptation are used. While both are acceptable, it is probably best to just use adaptation throughout or at least be consistent

4. Line 435 "both central and peripheral" but there is no discuss or evidence suggested for the central sources. Please discuss how a frequency-specific decline could occur in the central auditory system. 

5. Line 495-497: It is not clear how different patterns of synaptic activation may explain the contradictory effects of ketamine.

6. Line 695: It is not clear where the variability in the model arises from to require 20 trials each. Is it the Poisson process for the inputs? The stochastic conductance?

Reviewer #3: The authors explore a very important question in neurophysiology and their study sheds light on the complexities of the effect of anesthesia for electrophysiological recordings. The manuscript is clear and well written. The authors present sound data that supports their conclusions regarding the effect of Ketamine on high frequency tuned neurons and how this may affect the auditory processing of context dependent natural sounds.

Their implementation of the model using the known effects of ketamine and the differences that arise when comparing it to the real data provides convincing evidence that there are other effects to Ketamine that may not be widely understood. Here, for their set of experiments, adding the effect on high frequency components provides a model that reliably fits the data. Nevertheless, it opens the question of what other effects may not be evidenced because of the experimental paradigm. This work moves the field forward by demonstrating this effect and providing the groundwork for this to be tested against other experimental paradigms that may bring forward other unknown effects of anesthesia in neurophysiology.

I have only minor comments:

Line 80: suggested change of words "enable" instead of "allow"

Figure 6: the red and orange are very similar and the dark blue and lighter blue are very similar too. They are hard to see and I recommend the authors use colors that are more discriminable. Although clear in figure 6c, these small color differences are not explained in the figure legend, the authors should add this.

Line 486: The start of the sentence with the word "Besides" is confusing. I would advise the authors to reword.

---

## [Decision Letter · Decision Letter 2]

12 Jan 2023

Dear Dr Lopez Jury,

Thank you for your patience while we considered your revised manuscript "A neuron model with unbalanced synaptic weights explains asymmetric effects of ketamine in auditory cortex" for publication as a Research Article at PLOS Biology. This revised version of your manuscript has been evaluated by the PLOS Biology editors, the Academic Editor [and the original reviewers.

Based on the reviews, we are likely to accept this manuscript for publication, provided you satisfactorily address the data and other policy-related requests in their entirety. Please refer to these issues listed at the bottom of this email. 

***To ensure broad accessibility of this work, we also ask that you consider the following title change:

"A neuron model with unbalanced synaptic weights explains the asymmetric effects of anesthesia on the auditory cortex"

We expect to receive your revised manuscript within two weeks. 

*Published Peer Review History*

*Press*

Sincerely,

Kris

Kris Dickson, Ph.D., (she/her)

Neurosciences Senior Editor/Section Manager,

kdickson@plos.org,

PLOS Biology

DATA POLICY:

You may be aware of the PLOS Data Policy, which requires that all data be made available without restriction: http://journals.plos.org/plosbiology/s/data-availability. For more information, please also see this editorial: http://dx.doi.org/10.1371/journal.pbio.1001797 Note that we do not require all raw data. Rather, we ask that all individual quantitative observations that underlie the data summarized in the figures and results of your paper be made available. 

1) Thank you for providing supplemental files and access to the data at DOI: 10.12751/g-node.the3ge.We will need you to do a few minor modifications to the files provided as follows:

a) Please make sure that all data files are invariably referred to (in the manuscript, figure legends, and the Description field when uploading your files) using the following format verbatim: S1 Data, S2 Data, etc. Multiple panels of a single or even several figures can be included as multiple sheets in one excel file that is saved using exactly the following convention: S1_Data.xlsx (using an underscore).

b) Please check that the individual numerical values that underlie the summary data displayed in the following figure panels as they are essential for readers to assess your analysis and to reproduce it:

Fig1B; Fig2ABC graphs; Fig 3A heatmaps; Fig 4B-E; Fig6A,B; Fig7A-G; Fig9B-E; Fig10A-G; Fig11D

Supplemental Fig2A,B; Fig4B,D,D; Fig5B-F; Fig7A,B; Fig8A-F; Fig9

c) Please also ensure that figure legends **in your manuscript** include information on where the underlying data can be found, and ensure your supplemental data file/s has a legend.

DATA NOT SHOWN?

- Please note that per journal policy, we do not allow the mention of "data not shown", "personal communication", "manuscript in preparation" or other references to data that is not publicly available or contained within this manuscript. Please carefully check your submission for any such occurrences and either remove mention of these data or provide figures presenting the results and the data underlying the figure(s).

Reviewer remarks:

Reviewer's Responses to Questions

Do you want your identity to be public for this peer review?

Reviewer #1: No

Reviewer #2: Yes: Jordan D Chambers

Reviewer #3: No

Reviewer #1: I have read the revised manuscript. The authors have done an excellent job at revising the manuscript. I very much like the new Fig 1 and the rest of the changes made. I do not have additional comments.

Reviewer #2: The authors have answered all questions asked by the reviewers. They have added a significant amount of data and discussions to the manuscript in response to the reviewers questions. I believe the manuscript is now acceptable for publication.

Reviewer #3: The authors have addressed all my comments

---

## [Editor Report · Decision Letter 3]

27 Jan 2023

Dear Dr Lopez Jury,

Thank you for the submission of your revised Research Article "A neuron model with unbalanced synaptic weights explains the asymmetric effects of anaesthesia on the auditory cortex" for publication in PLOS Biology. On behalf of my colleagues and the Academic Editor, Manuel Malmierca, I am pleased to say that we can in principle accept your manuscript for publication, provided you address any remaining formatting and reporting issues. These will be detailed in an email you should receive within 2-3 business days from our colleagues in the journal operations team; no action is required from you until then. Please note that we will not be able to formally accept your manuscript and schedule it for publication until you have completed any requested changes.

PRESS

Sincerely, 

Kris

Kris Dickson, Ph.D., (she/her)

Neurosciences Senior Editor/Section Manager

PLOS Biology

kdickson@plos.org